# Volcanic emission estimates from the inversion of ACTRIS lidar observations and their use for quantitative dispersion modelling

Anna Kampouri[1,2], Vassilis Amiridis[1], Thanasis Georgiou[1,3], Stavros Solomos[4], Anna Gialitaki[1,5], Maria Tsichla[1,6], Michael Rennie[7], Simona Scollo[8], and Prodromos Zanis[2]

[1]Institute for Astronomy, Astrophysics, Space Applications and Remote Sensing, (IAASARS) National Observatory of Athens, 10560 Athens, Greece.
[2]Department of Meteorology and Climatology, School of Geology, Aristotle University of Thessaloniki, 54124 Thessaloniki, Greece.
[3]Laboratory of Atmospheric Physics, Aristotle University of Thessaloniki, 54124 Thessaloniki, Greece.
[4]Research Centre for Atmospheric Physics and Climatology, Academy of Athens, 10680 Athens, Greece.
[5]Department of Physics and Astronomy, Earth Observation Science Group, University of Leicester, Leicester, UK
[6]Environmental Chemical Processes Laboratory, Department of Chemistry, University of Crete, Greece.
[7]European Centre for Medium-Range Weather Forecasts, Reading RG2 9AX, UK.
[8]Istituto Nazionale di Geofisica e Vulcanologia, Osservatorio Etneo, 95125 Catania, Italy.

Correspondence to: Anna Kampouri (akampouri@noa.gr)

**Abstract.** Modeling the dispersion of volcanic particles following explosive eruptions is critical for aviation safety. To constrain the dispersion of volcanic plumes and assess hazards, calculations rely on accurate characterization of the eruptions source term, e.g., variation of emission rate and column height with time and the prevailing wind fields. This study introduces an inverse modeling framework that integrates a Lagrangian dispersion model with lidar observations to estimate emission rates of volcanic particles released during an Etna eruption. The methodology consists of using the FLEXPART model to generate source-receptor relationships (SRR) between the volcano and a lidar system that observed the volcanic plume. These SRR are then used to derive the emission rates based on observational data including volcanic ash plume heights from INGV-EO and Polly[XT] lidar retrievals. We leverage data from the ACTRIS Polly[XT] lidar that operates at the PANhellenic GEophysical observatory of Antikythera (PANGEA-NOA). The inversion algorithm utilizes lidar observations and an empirical a-priori emission profile to estimate the volcanic particle source strength, accounting for altitude and time of the plume's evolution. Additionally, to study the impact the wind fields have on volcanic ash forecasting, the experiment is repeated using fields that assimilate Aeolus wind lidar data. Our approach applied to the 12 March 2021 Etna eruption, accurately captures a dense aerosol layer between 8 and 12 km above PANGEA-NOA station. Results show a minimal difference of the order of 2 % between the observed and the simulated ash concentrations. Furthermore, the structure of the a-posteriori ash plume closely resembles the ash cloud image captured by the SEVIRI satellite above Antikythera island, highlighting the novelty of the inversion results. The presented inversion algorithm coupled with Aeolus data, optimizes both the vertical emission distribution and Etna emission rates, advancing our understanding and preparedness for volcanic events.

# 1 Introduction

Volcanic ash constitutes a significant hazard to aviation when it is emitted at aircraft cruising altitudes (9 - 11 km), with potential consequences including aircraft engine failure (Guffanti et al., 2005), inaccurate readings of critical navigational instruments, and reduced visibility due to external aircraft corrosion (Clarkson and Simpson, 2017; ICAO, 2016).

In the case of a volcanic eruption, urgent decisions are necessary to determine safe flight routes and ensure that airborne aircraft land safely. While safety remains the top priority, the grounding and rerouting of flights leads to large financial losses e.g., the

2010 eruption of Eyjafjallajökull in Iceland reportedly cost the airline industry over 1 billion USD (Mazzocchi et al., 2010; Oxford Economics, 2012).

Information on volcanic ash dispersion after an eruption is provided to operators by specialized early warning systems (EWSs) operated by the Volcanic Ash Advisory Centres (VAACs) (Fearnley et al., 2018). These systems are typically relied on deterministic volcanic ash transport and dispersion models (VATDM), to offer short-term forecasts of the volcanic ash cloud.

Although VAACs specify the expected location of the ash cloud, usually they do not provide quantitative information about ash concentration. In the spotlight of the expected rise in the number of flights over volcanically active regions in the near future (as indicated by EUROCONTROL, 2022), the probability of encountering volcanic ash at aircraft cruising altitudes will proportionally increase. Consequently, the challenge is to minimize uncertainties in short-term forecasts of volcanic ash dispersion.

The primary sources of uncertainties in deterministic transport models originate from the eruption source parameters, the various model parameterizations (such as wet deposition), and the driving meteorological conditions (Dacre et al., 2011; Prata and Lynch, 2019; Stohl et al., 2011). Typically, VATDMs require specification of parameters about the volcanic events, including a vertical profile of ash emission rates, particle size distribution, and the ash density (Harvey et al., 2020).The eruption start time can be estimated through satellite observations or by local Volcano Observatories. Various remote sensing

techniques exist to estimate the height of the ash plume (Petersen et al., 2011). Though it should be mentioned that information that rely on observations from passive sensors practically have limited sensitivity to the ash layer height. Mass eruption rates are typically evaluated using empirical relationships based on observed plume heights (Mastin et al., 2009). However, these empirical relationships often fail to consider secondary factors influencing plume height, such as meteorological conditions. The long-range transport of volcanic particles is influenced by tropospheric and/or stratospheric winds, and particularly the

vertical wind shear, which is frequently inaccurately represented in numerous Numerical Weather Prediction (NWP) models (Harvey et al., 2020; Houchi et al., 2010; Stoffelen et al., 2020).

Moreover, volcanic particles can influence the planetary radiative balance through both direct and indirect effects, introducing significant uncertainties in plume dispersion and lifetime. The direct effect involves the scattering and absorption of solar and terrestrial radiation, where fine ash and sulfate aerosols contribute to surface cooling or atmospheric warming depending on

particle composition, size distribution, and injection plume height (Robock, 2000; Sicard et al., 2025). The indirect effect relates to the role of volcanic particles in cloud micro- and macro-physical properties. Ash particles can act as cloud

condensation nuclei (CCN), facilitating water droplet formation and, under specific pressure and temperature conditions, as ice nuclei (IN) (Guerrieri et al., 2023). These processes can alter cloud optical and microphysical properties, enhance cloud reflectivity, affect cloud lifetimes and increase the uncertainties in radiative transfer. Additionally, volcanic ice clouds can hide possible ash layers and pose a severe threat to aviation safety. Atmospheric transport models often struggle to account for these complex interactions, leading to uncertainties in plume evolution, trajectory forecasts, and deposition estimates. Furthermore, the absence of significant physical processes and dependence on empirical relations and data from previous eruptions further contributes to substantial uncertainties in estimates of the erupted mass.

Over the past two decades, significant progress has been made in integrating remote sensing data into atmospheric transport models to enhance the forecasting of volcanic emissions and their dispersion. Satellite observations from both polar orbiting and geosynchronous thermal infrared instruments have been used to retrieve ash mass loadings (Clarisse et al., 2010; Pavolonis et al., 2013; Prata and Prata, 2012). Additional sensors, including Moderate Resolution Imaging Spectrometer (MODIS), Second Generation Spin-stabilised Enhanced Visible and Infra-Red Imager (SEVIRI), Atmospheric Infra-Red Sounder (AIRS), Ozone Monitoring Instrument (OMI), Multi-angle Imaging Spectroradiometer (MISR), and CALIOP lidar on board of the CALIPSO have provide valuable data in volcanic ash detection and retrievals (Eckhardt et al., 2008; Francis et al., 2012). A comprehensive discussion on the application of satellite remote sensing for volcanic ash monitoring in aviation hazard mitigation is provided by Prata, (2009).

In addition, ground-based lidar networks, such as the European Aerosol Research Lidar Network (EARLINET), have played a crucial role in validating the accuracy of transport model outputs and improving dispersion simulations, by providing high-resolution vertical profiles of volcanic aerosols (Pappalardo et al., 2004).

Advancements in atmospheric transport and dispersion modeling have further facilitated the integration of these observational datasets. Models like the Numerical Atmospheric-dispersion Modelling Environment (NAME; Jones et al., 2007) which is used operationally by the London Volcanic Ash Advisory Centre (LVAAC), the Hybrid Single-Particle Lagrangian Integrated Trajectory model (HYSPLIT) (Stein et al., 2015) and the FLEXible PARTicle dispersion model (FLEXPART) (Eckhardt et al., 2008; Kristiansen et al., 2010, 2012, 2014; Stohl et al., 2011) have been extensively used for volcanic ash forecasting, often incorporating satellite and lidar data to refine model inputs and improve predictive accuracy.

The integration of remote sensing data into atmospheric transport models has been significantly advanced through inversion algorithms. In previous studies (Eckhardt et al., 2008; Kristiansen et al., 2010), inversion algorithms were developed using satellite column retrievals and tested to estimate the vertical distribution of sulphur dioxide ($SO_2$) emission rates for quasi-instantaneous volcanic eruptions such as the 2007 Jebel at Tair and the 2008 Kasatochi eruptions. Seibert et al., (2011) examined the uncertainties of the various configurations for the 2008 Kasatochi case study and expanded the method to estimate the uncertainty of the retrieved source emissions (a-posteriori uncertainties).

The inversion algorithm was further used by Stohl et al., (2011) for volcanic ash emission rates as a function of altitude and time. While Kristiansen et al., (2012) improved the volcanic ash inversion techniques using various inputs to better constrain the 2010 Eyjafjallajökull eruption.

In Amiridis et al., (2023), it is demonstrated that volcanic ash early warning systems can be significantly enhanced from the assimilation of Aeolus wind fields. Notably, these improvements are most pronounced over under-sampled geographical regions, such as the Mediterranean Sea, as volcanoes are often situated in remote areas lacking surface-based observation networks. Moreover, the study indicates that the positive effect of Aeolus wind data assimilation is more pronounced in the middle and upper troposphere (mostly between 7 and 15 km), compared to the lower troposphere. This may highlight under-sampling issues, since the in situ observations (like radiosondes) traditionally used for data assimilation, exhibit lower vertical resolution in the upper troposphere (Rennie et al., 2021). Considering that volcanic ash plumes are typically injected in upper-tropospheric and lower-stratospheric heights, their transport is largely influenced by upper tropospheric winds hence accuracy in dispersion modelling is advanced from high accuracy wind fields assimilation.

Building on these advancements, our study further investigates improvements in ash emission estimations by developing an inversion method that integrates Aeolus wind data, ground-based lidar observations, and transport model simulations. This approach aims to enhance the accuracy of volcanic emission source terms and reduce uncertainties in dispersion forecasting. We specifically focus on the Etna eruption that occurred on 12 March 2021, coinciding with the investigations provided by Amiridis et al., 2023 and Kampouri et al., 2023. During this event, Aeolus had a close overpass to Etna, providing valuable observations around the volcano. Additionally, the transported volcanic plume was captured in the region of in the Eastern Mediterranean by the ground-based Polly$^{XT}$ lidar system of the PANhellenic GEophysical observatory of Antikythera (PANGEA-NOA) island, in Greece downwind of Etna volcano. This allows for direct comparisons of observations against forecasts, with and without assimilation of Aeolus data, denoted as "w" and "w/o" Aeolus, respectively (as indicated in the studies by Amiridis et al., 2023; Kampouri et al., 2023).

## 2 The Case of 12 March–14 March 2021 Etna Volcanic Eruption

### 2.1 Volcanic Activity

Mt. Etna in Italy, recognized as one of the most active volcanoes on Earth, has undergone significant volcanic activity, particularly since February 2021. During this period, the stratovolcano experienced numerous paroxysmal episodes, leading to frequent tephra and $SO_2$ emissions. A notable event occurred on 12 March 2021, marking one of the most powerful lava fountain episodes observed at the South East Crater since 2020 (Calvari et al., 2021). The volcanic activity started with Strombolian-type eruptions around 02:35 UTC, escalating in both frequency and intensity until 07:35 UTC, when surveillance cameras from the Istituto Nazionale di Geofisica e Vulcanologia, Osservatorio Etneo (INGV-OE) (Corradini et al., 2018; Scollo et al., 2019), captured the formation of a sustained lava fountain.

Throughout the paroxysmal phase, the eruptive column gradually reached a height up to 9 km a.s.l. (Figure 1). The variation in the eruption column was detected by the visual surveillance camera at the CUAD in Catania (ECV) calibrated by the INGV-OE (Figure 1). The volcanic plume drifted eastwards under the influence of prevailing westerly winds dominant in the eastern Mediterranean region at the time. According to the Volcano Observatory Notice for Aviation (VONA) messages, the INGV-

EO observatory (INGV-EO; Corradini et al., 2018; Scollo et al., 2019) issued a RED warning alert, from 06:18 to 08:44 UTC, on 12 March 2021, when the strongest ash emission was observed, while an ORANGE alert was issued at 12:30 UTC when

the lava fountain ceased, and the volcanic ash plume was dispersed in the atmosphere (Calvari et al., 2021). Additionally, the eruptive activity resulted in abundant tephra fallout, covering several towns on the east flank of the volcano crater, and a lava flow field expanding on the east and north-east flank. In this study, the cloud heights reported by VONA are used as a-priori information to initialize the volcanic ash dispersion simulations, conducted with the FLEXPART (flexible particle dispersion) Lagrangian model (Brioude et al., 2013; Pisso et al., 2019; Stohl et al., 2005). The FLEXPART ash transport model is driven

by wind fields simulated by the WRF regional meteorological model (version 4) (Skamarock et al., 2019), which, in turn, derives initial and boundary conditions from the ECMWF-Integrate Forecast System (IFS) (ECMWF, 2021) global model (for additional information see Sect. 3.3).

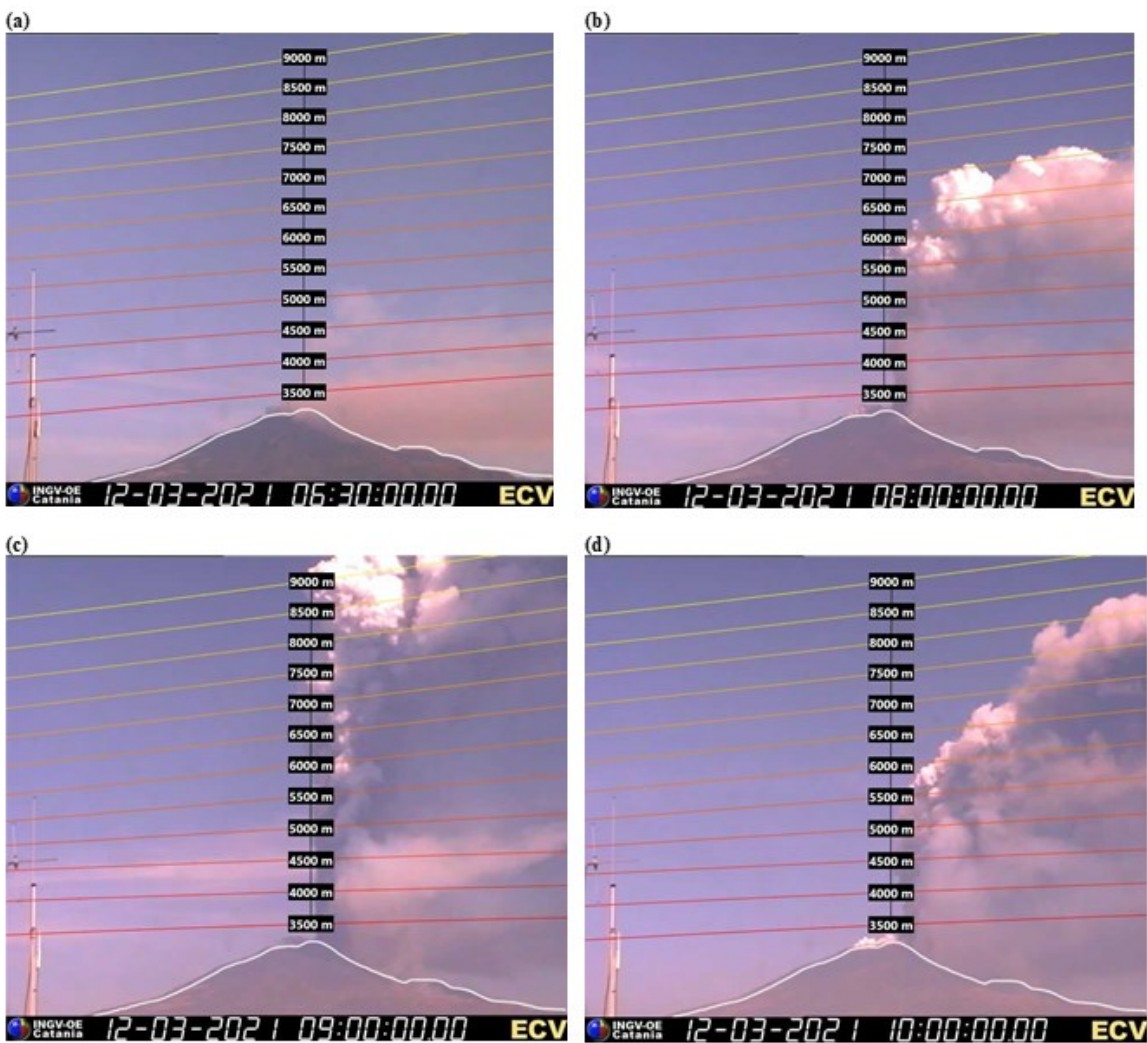

**Figure 1: Etna activity on 12 March 2021 as seen from INGV-OE. Ash plume images from ECV calibrated camera monitored the explosive volcanic activity between 5 and up to 9 km a.s.l. a) weak ash plume at 06:30 UTC, with an upper part aligning more vertically; b) strong vertical plume at 08:00 UTC, shifted eastward; c) strong ash plume at 09:00 UTC, with a lower and more diluted cloud caused by the lava flow expanding eastward and d) decrease of the explosive activity after 10:00 UTC (figures are taken from the INGV-OE automatic system described in Corradini et al., 2018 and Scollo et al., 2019).**

## 3 Methods and Data

The inverse method employed in this study to estimate volcanic ash emissions integrates a-priori information on ash emissions, ground-based lidar observations, and simulations with a dispersion model, resulting in improved ash emission estimates. Figure A 1, presents a schematic workflow outlining the methodology followed in this study, providing a clear overview of the steps involved in our approach. In this section, we describe the datasets and methods employed in the inverse modeling process.

### 3.1 PANGEA-NOA ground-based data (Lidar-Polly<sup>XT</sup>)

The PANGEA-NOA observatory established its first operations in June 2018 in the remote island of Antikythera, Greece. The atmospheric circulation pattern at PANGEA-NOA favours the transport of air masses carrying an abundance of different aerosol types such as windblown Sahara dust, Etna volcanic aerosols, smoke from wildfires and anthropogenic pollution from major cities. Hence, this coastal site constitutes an ideal place to study natural aerosols under the prevailing background conditions of the Eastern Mediterranean.

The Mediterranean region, particularly its Eastern basin, serves as a confluence of air masses originating from Europe, Asia, and Africa. In this region, anthropogenic emissions from large urban centers interact with natural emissions from the Saharan and Middle Eastern deserts, smoke from frequent wildfires, and volcanic particles from eruptions, notably from Mt. Etna and Icelandic volcanoes. Additionally, the atmosphere over the Eastern Mediterranean contains background marine aerosols and pollen particles from oceanic and vegetative sources. Aerosols exert a variety of effects on regional weather and climate,

impacting solar radiation, visibility, and human health, and they pose significant concerns for aviation safety (WMO, 2024). The Eastern Mediterranean is characterized by a Mediterranean climate, with hot, dry summers and mild, wet winters. This seasonal variability is driven primarily by the interaction between mid-latitude westerlies and subtropical high-pressure systems (Lensky et al., 2018). During winter, the region experiences frequent passage of extratropical cyclones originating from the North Atlantic and Mediterranean storm tracks, bringing precipitation and colder temperatures. In contrast, summer

conditions are dominated by the expansion of the subtropical height, leading to stable atmospheric conditions and minimal rainfall (ECMWF, 2010).

Synoptic-scale circulation in the Eastern Mediterranean plays a crucial role in shaping weather patterns and atmospheric dynamics. The atmospheric circulation over the eastern Mediterranean is dominated by persistent northerly and westerly winds, favoring the advection of volcanic products from Etna to Greece (Kampouri et al., 2020; Scollo et al., 2013). Research has

identified several dominant synoptic types that influence the region, including cyclonic systems, anticyclonic patterns, and blocking heights (Rousi et al., 2014). These circulation patterns significantly impact the transport of aerosols, moisture, and pollutants, affecting regional air quality and climate variability. Furthermore, the region's proximity to large-scale circulation features such as the subtropical jet stream and the African monsoon system contributes to complex seasonal interactions (Lensky et al., 2018).

Currently, a Polly<sup>XT</sup> lidar system (Baars et al., 2017; Engelmann et al., 2016) and a sun/sky-photometer of CIMEL Electronique (Giles et al., 2019; Goloub et al., 2007) operate continuously at PANGEA-NOA to provide profiles and columnar aerosol properties with high accuracy and resolution.

Polly<sup>XT</sup> is a multi-wavelength, Raman, polarization lidar with 24/7 remote operation capability. The system operates in 355, 532 and 1064 nm and is equipped with 12 detectors to measure light elastically and in-elastically (at 387, 407 and 607 nm)

backscattered from atmospheric constituents. Polarization capability also enables the detection and vertical separation of non-spherical (e.g., volcanic ash, dust) from spherical aerosols (e.g., smoke, pollution, marine particles).

The CIMEL sun/sky-photometer measures direct solar and sky radiance at several wavelengths (340, 380, 440, 500, 675, 870, 1020 and 1640 nm), to derive column integrated aerosol optical and microphysical properties (Dubovik et al., 2006).

Observations from both sensors are of strong interest for Pan-European and global networks such as the Aerosol, Clouds and Trace Gases Research Infrastructure (ACTRIS-RI), the European Aerosol Research Lidar Network (EARLINET) and the global AErosol RObotic NETwork (AERONET: https://aeronet.gsfc.nasa.gov/); in all of which measurements taken at PANGEA-NOA are submitted on a regular basis.

### 3.1.1 Ash mass calculation using remote sensing data

Volcanic ash mass estimates were derived from a combination of Polly$^{XT}$ lidar measurements and sun-photometer observations. First, the lidar measurements were averaged over the 3-hour period when the volcanic layer was observed above Antikythera, and the standardized EARLINET algorithm Single Calculus Chain (SCC) (D'Amico et al., 2015), was used to derive the particle backscatter coefficient ($\beta_p$) and particle linear depolarization ratio ($\delta_p$) profiles.

These profiles were then used to disentangle the contribution of large, non-spherical ash particles to the observed volcanic plume and then calculate the ash mass concentration with the "POlarization-LIdar PHOtometer Networking" (POLIPHON) method (Ansmann et al., 2012; Mamouri and Ansmann, 2017), tailored for Etna ash as described in Kampouri et al. (2020). More specifically, the following equation was used:

$$m_a = \rho_a * c_{v,a}(\lambda) * \beta_{p,a}(h, \lambda) * S_{p,a}(h, \lambda), \tag{1}$$

where $m$ is the mass concentration, $a$ indicates an aerosol type, $\rho$ represents the particle mass density (for volcanic ash particles is 2.6±0.6 g/cm$^3$ following the study of Ansmann et al., 2011), $\lambda$ is the wavelength, $c_v(\lambda)$ is the so-called volume to extinction conversion factor, derived from sun-photometer measurements, h is the height above ground and $S_p(\lambda, h)$ is the ratio of the particle extinction to particle backscatter coefficient (lidar ratio).

As $m_a$ calculation is sensitive to the aerosol type, under simultaneous presence of multiple aerosol components in the atmospheric column, a decomposition of the total particle backscatter coefficient $\beta_{p,}$ is needed prior to the mass concentration calculation. In POLIPHON, this decomposition is supported for up to two aerosol types, one exhibiting large particle depolarization ratio values (usually dust or volcanic ash) and one that does not (marine, continental or tropospheric smoke and their mixtures). To separate the contribution of the depolarizing ($\beta_{p,d}(h, \lambda)$) and the non-depolarizing ($\beta_{p,nd}(h, \lambda)$) aerosol component to the total particle backscatter coefficient, we apply the following equations:

$$\beta_{p,d}(h, \lambda) = \beta_p(h, \lambda) \frac{\left(\delta_p(h, \lambda) - \delta_{p,nd}(h, \lambda)\right)\left(1 + \delta_{p,d}(h, \lambda)\right)}{\left(\delta_{p,d}(h, \lambda) - \delta_{p,nd}(h, \lambda)\right)\left(1 + \delta_p(h, \lambda)\right)}, \tag{2}$$

$$\beta_{p,nd}(h, \lambda) = \beta_p(h, \lambda) - \beta_{p,d}(h, \lambda), \tag{3}$$

Polly$^{XT}$ lidar signals are sensitive to aerosol particles in the radius range from about 50 nm to a few micrometers (Weitkamp, 2005). For FLEXPART, the size range considered for volcanic ash particles is between 5 and 21 μm in diameter, and thus within the range that is detectable from Polly$^{XT}$. Uncertainties in the ash mass concentration calculation using the POLIPHON method, rise from the input parameters errors that propagate into Eq. (1) and are expected to be in the order of ~40 % (Ansmann et al., 2011b). The technique has been validated against synergistic retrievals that combine multi-wavelength lidar and sun/sky-radiometer observations (sensitive up to 15 μm in particle radius (Lopatin et al., 2013, 2021) for dust and volcanic ash particles and has been found to perform well (Konsta et al., 2021; Wagner et al., 2013).

In Table 1, we summarize the values and uncertainties of parameters used as input for the above. The lidar ratio of coarse mode volcanic ash at 532 nm is reported to range between 40 and 60 sr in the literature (see for example Groß et al., 2012; table 3 for particle extinction and backscatter values in Floutsi et al., 2023 and Gasteiger et al., 2011). For the fine mode aerosols, we use a mean value of 60 sr following the values reported in the literature for particles of sulphuring nature (see for example Floutsi et al., 2023; Müller et al., 2007). We also account for a lidar ratio retrieval uncertainty of ~30% to capture the measurement range (Ansmann et al., 2012; Giannakaki et al., 2015; Groß et al., 2013). The particle density values $\rho$ follow from OPAC model for coarse mode mineral component and water soluble component for ash and sulfate particles respectively (Hess et al., 1998; Koepke et al., 2015). For the water soluble component, we assume values at relative humidity of 0% which is considered representative for the altitudes of the volcanic layers. The coarse mode component is not considered as hudrophylic. Finally, the extinction to mass conversion factors cv are taken from Ansmann et al., (2011a) for ash and fine mode particles respectively.

Table 1: Parameters used for lidar profiles decomposition and mass concentration calculation.

|  | $\rho_a$ [μm cm$^{-3}$] | $c_{v,a,532nm}$ | $\delta_{p,a,532nm}(h)$ | $S_{p,a,532nm}(h)$[sr] |
|---|---|---|---|---|
| Ash particles | $2.6 \pm 0.6$ | $0.6 \pm 0.1$ | $0.36 \pm 0.02$ | $50 \pm 10$ |
| Sulfates | $1.5 \pm 0.3$ | $0.18 \pm 0.02$ | $0.05 \pm 0.01$ | $60 \pm 20$ |

## 3.2 Aeolus high spectral resolution lidar (HSRL) data

Aeolus, the European Space Agency's (ESA) wind mission, carried the world's first high spectral resolution Doppler wind lidar (HSRL) in space (Stoffelen et al., 2006; Straume-Lindner et al., 2021). Launched in August 2018, Aeolus's aim was to retrieve horizontal wind profiles in the troposphere and lower stratosphere. The mission's primary objective was to showcase this innovative technology in space to enhance weather forecasts and to advance our understanding of atmospheric dynamics, particularly in the tropics. Additionally, Aeolus aimed to contribute valuable insights into the intricate interactions between the atmospheric constituents, water cycles, and the broader climate system (Rennie et al., 2021; Straume-Lindner et al., 2021). Aeolus wind data demonstrated notable quality and coverage, leading to substantial enhancements in NWP forecasts, particularly within the tropics and Southern Hemisphere. The improvement in wind forecasts ranges from 0.5 % to 2 %, in terms of root-mean-square error, maintaining a significant impact even into the medium range weather forecasting. The most

substantial impact was observed at approximately 100 hPa in the tropics, particularly over the east Pacific Ocean. This is attributed, in part, to the tropics having a relatively limited coverage of high-quality radiosonde wind profiles. Additionally, the wind field in the tropics is less constrained by temperature information from other satellites (Rennie et al., 2021). Furthermore, Aeolus had the capability to retrieve aerosol and cloud profiles, offering valuable data for assimilation or evaluation in volcanic ash dispersion modeling. It is essential to note, however, that these retrievals face limitations due to the absence of a dedicated lidar channel for detecting cross-polarized light (with respect to the emitted radiation) returns. This absence is particularly crucial for capturing the backscattered light from non-spherical particles like volcanic ash. Consequently, caution is advised when utilizing Aeolus observations in such cases. Despite this limitation, the Aeolus mission demonstrated its efficacy in enhancing wind forecasts, particularly over under-sampled regions, such as the tropics (Rennie et al., 2021). Similarly, Aeolus can be used over under-sampled remote areas with active volcanoes, contributing to improved simulations of volcanic ash dispersion following eruptions.

### 3.3 FLEXPART-WRF model setup

To perform meteorological simulations over the study region of the Eastern Mediterranean the Advanced Research WRF model version 4 (Skamarock et al., 2019) is used. The spatial resolution of the model is 12 × 12 km for a total of 351 × 252 grid points, and 31 vertical levels (up to 50 hPa). The simulation period starts on 12 March 2021, at 00:00 UTC (six hours earlier than the FLEXPART runs, to accommodate for the model's 12h spin-up) and ends on 14 March 2021, at 18:00 UTC, with hourly outputs. Table 2 summarizes the Physics Parameterizations (PP) schemes for the WRF-ARW simulations.

In the context of this study, two versions (ECMWF, 2021) of the initial and boundary condition fields from the IFS were utilized. These fields, provided at a spatial resolution of 0.125° × 0.125°, with 137 vertical model levels, serve as inputs for the WRF-ARW regional model. One version incorporates assimilated Aeolus Rayleigh-clear and Mie cloudy horizontal line-of-sight (HLOS) L2B wind profiles (referred to as the "w" Aeolus experiment), while the other version is without Aeolus data (referred to as the "w/o" Aeolus experiment). The initial conditions without Aeolus assimilation adhere to the model setup utilized in the Observing System Experiments (OSEs) conducted by Stoffelen et al., (2006).

The WRF-ARW runs rely on initial and boundary conditions generated from ECMWF-IFS, with boundary conditions updated at 6-hour intervals. Sea Surface Temperature (SST) analysis data, obtained from the Copernicus Marine Environment Monitoring Service (CMEMS) at a spatial resolution of 1/12°, supplement these simulations. The WRF-ARW model configuration utilized in this study is consistent with that employed in the study of Amiridis et al. (2023).

The volcanic ash plume transport simulations were done with the Lagrangian particle dispersion model FLEXPART (Brioude et al., 2013; Pisso et al., 2019; Stohl et al., 2005) in a forward mode. These simulations rely on hourly meteorological fields from the WRF-ARW model, initiated with IFS datasets. The use of 1-hourly WRF meteorological fields at a 12 × 12 km spatial resolution allows for a more detailed representation of the volcanic plume dispersion. The initial simulations, in which we used an a-priori emission profile for the eruption emissions taken from VONA alerts (from now on referred to as 'a-priori volcanic ash plume transport'), were initiated at the reported start time of the eruption 07:00 UTC on 12 March 2021 and were completed

at 00:00 UTC on 14 March 2021 with a total of 100,000 particles released in each forecast. The model layers were divided into 18 layers with 1km vertical resolution, in the range extending from 1 to 18 km above ground level (a.g.l.) We estimate the a-priori mass eruption rate (MER) for ash particles following Mastin et al., 2009 and Scollo et al., 2019, by inverting the observed plume heights over the Etna summit crater from VONA reports and field observations as observed by the INGV observatory, using the 1-D plume model of Degruyter and Bonadonna, (2012). The initial injection height in the model is set to the altitude of the Etna summit craters (3.3 km a.s.l) up to 9 km a.s.l., based on the VONA reports (Corradini et al., 2018; Scollo et al., 2019) and field observations. Also, the gravitational particle settling (Näslund and Thaning, 1991) was determined assuming spherical particles with a density of 2450 kg/m$^3$. The particle density value used in the FLEXPART model differs slightly from the density used in Table 1 (2.6 ± 0.6 g/cm³) due to differences in shape assumptions, size distributions, and literature sources referenced in various calculations. The size distribution of volcanic ash particles was described using four size bins (3, 5, 9, and 21 μm in diameter), as these cover the size distribution relevant for long-range transport (≤ 25 μm diameter) (Beckett et al., 2022; Dacre et al., 2011; Durant et al., 2010).

To derive the source–receptor relationships (SRR), the FLEXPART-WRF model was used once again in a forward mode (see Appendix A, Figure A 2), considering the same four ash size bins as those used in the a-priori volcanic ash plume transport. The SRR model data, which represent all potential dispersion scenarios of the ash plume, are compared with the lidar retrievals at PANGEA-NOA. For each grid point in the considered domain, FLEXPART ash column loadings released from one particular emission time and height are matched with the corresponding time and grid point of the lidar ash mass retrieval.

FLEXPART-SRR were driven by the same hourly meteorological fields from the WRF-ARW model, utilizing both control and assimilated datasets (ECMWF, 2021) to quantitatively evaluate the impact of data assimilation. Subsequently, these SRRs were used to initialize the inversion algorithm, constrained with the Polly[XT] ground-based lidar measurements of volcanic particles.

It was assumed that the ash emissions occurred between the ground and 16 km a.g.l. over the Etna volcano. The total height range was discretized into 79 layers of 200 m thickness. For each layer, 150,000 unit mass particle traces were uniformly released along a vertical line source every two hours (from 04:00 to 06:00 UTC until 12:00 to 14:00 UTC). Additionally, the model layers were divided into 74; 70 layers between 200 m and 14 km, with a vertical resolution of 200 m, 3 layers between 14 and 16 km a.g.l., (per 1 km) and another layer from 22 to 50 km a.g.l. These model-derived column values represent source-receptor relationships, since they were obtained with a unit mass as source. The actual mass released at each level is determined through the inversion. Following the inversion, a single longer 'posteriori' simulation over the period 12 to 14 March 2021 was made releasing 200,000 particles according to the estimated emission profile. The output from this simulation was produced at the same vertical and horizontal resolution as the a-priori FLEXPART simulation.

**Table 2: Configuration of the PP schemes for the WRF-ARW simulations.**

| PP | Schemes | References |
| --- | --- | --- |

| Microphysics (MP) | Thompson | (Thompson et al., 2008) |
|---|---|---|
| Surface Layer (SFL) | Monin-Obukhov (Janjic Eta) | (Janjic, 2002) |
| Planetary Boundary layer (PBL) | Mellor-Yamada-Janjic (MYJ) | (Janjic, 2003) |
| Cumulus Parameterization (CUM) | Tiedtke | (Zhang et al., 2011) |
| Longwave & Shortwave Radiation (RAD) | Rapid Radiative Transfer Model (RRTMG) | (Iacono, et al., 2008) |
| Land Surface (LSM) | NOAH | (Chen, F. and Dudhia, J., 2001) |

### 3.4 Inversion algorithm

The inversion method employed here for ash source estimations is based on a cost function minimization approach. Similar work has been done by Eckhardt et al., 2008; Kristiansen et al., 2010 and Stohl et al., 2011. In these studies, an inversion algorithm was developed to calculate the vertical distribution of sulphur dioxide and ash emission rates for instantaneous volcanic eruptions. Satellite retrievals, typically of ash column loading, have been combined, in those analyses, with VATDM simulations using inversion techniques to provide time-evolving estimates of these significant quantities.

In satellite retrieval techniques, numerous advantages exist where estimates of ash cloud top height and ash column loading are typically available (Francis et al., 2012; Pavolonis et al., 2013). Additionally, MER can be estimated through empirical relationships under specific assumptions, which are especially useful when satellite images are unavailable or limited, such as during the early stages of an eruption (Pouget et al., 2013; Prata et al., 2022). However, direct retrievals of the vertical distribution within the eruption column are not feasible. Ground-based and airborne radar observations, which are sensitive to larger particles and can penetrate optically thick plumes, provide a complementary source of information to retrieve near-source plume properties such as mass eruption rate and column height.

The present study brings together: i) the inverse modeling by initiating the inversion simulations with mass concentrations derived from ground-based lidar observations downwind, combined with the source-receptor relationships calculated from the FLEXPART-WRF model, and ii) the integration of Aeolus meteorological wind fields (ECMWF, 2021) into the FLEXPART-WRF model (for more details see Sect. 3.3). The overarching goal is to optimize both the vertical emission distribution, and the ash emission rates near the source, following the volcanic eruption. From the inversion scheme a total ash emission profile of the eruption is obtained, which can be utilized to generate robust ash forecasts constrained by lidar observations.

We perform the inversion using a Bayesian approach to provide the best estimate of the emissions profile for fine ash (with particles 3, 5, 9, and 21 μm in diameter) that can be transported over long distances. We follow the general concept of source-receptor relationships (Seibert and Frank, 2004), where the relations between each measurement and a potential source of the emission is calculated (here using FLEXPART-WRF) and stored as the source-receptor matrix (SRM) for each vertical level and for four ash size bins (as described in Sect. 3.3). The n=79 unknowns (source elements) are put into a state vector $x$, while the $m$ observed values are put into a vector $y_o$, where the subscript 'o' stands for the Polly[XT] lidar observations. Then, the state vector can be calculated from the inversion of a forward model M that connects $y_o$ and $x$ as follows:

$$y_o = M(x) + e_y, \tag{4}$$

implying a linear relationship in which $y_o$ is a vector of spatiotemporal lidar measurements, $M$ is the $n \times m$ SRM calculated by FLEXPART-WRF, describing the sensitivity of each observation to a unit release rate, $e_y$ represents lidar measurement errors (which are not accounted for in the algorithm) and $x$ is the ash emission vector to be estimated. $M(x)$ is equivalent to running a VATDM with $x$ as the input release profile. Since $M$ is calculated using such a model, it inherits the biases that are inevitable in VATDMs. As a result, it may diverge from the true dispersion and may not necessarily align with the observations on the left-hand side of Eq. (4), even if is the true release profile (Fang et al., 2022). Given that the problem is underdetermined, the solution of the linear inverse problem in Eq. (4) is not straightforward and further assumptions are needed.

The most common are assumptions imposed on the unknown emission vector $x$ such as non-negativity of its elements, smoothness of the emission (Fang et al., 2022) or measurement/emission sparsity (Li et al., 2018), e.g., the assumption that the emission element remains zero unless other evidence is present in measured and modeled data. Under these assumptions, the problem in Eq. (4) can be solved by minimizing the distance between the left and the right sides of the equation. To enhance the stability of the inversion outcome, a-priori emissions are also used, representing our best estimate of $x$ before the observations are made (see in Sect. 3.4.1). Including an explicit a-priori source vector $x^a$, we can express the equation as follows:

$$M(x - x^a) \approx y_o - Mx^a, \tag{5}$$

and as an abbreviation

$$M\tilde{x} \approx \tilde{y}. \tag{6}$$

The inversion scheme presented here is done by minimizing a cost function $C$, which comprises the following system of equations:

$$C_1 = y_o - M^T x, \tag{7}$$

$$C_2 = x - x^a, \tag{8}$$

$$C_3 = \epsilon Dx, \tag{9}$$

$C_1$ quantifies the difference between the modeled data and the observations, $C_2$ the deviation from the a-priori estimations, and $C_3$ imposes a smoothness regularization term.

The cost function $C$ first calculates the misfit $C_1$ between the profiles at the receptor points, as observed by the lidar ($y_o$) and the data as modelled by FLEXPART ($M^T x$).

The second term $C_2$ (Eq. (8)) accounts for the difference, between the a-posteriori estimates of the emission rates $x$ and the a-priori estimates $x^a$ (for details on the calculation of the a-priori vector see Sect. 3.4.1). To enforce smoothness in the vertical profile of emissions, a regularization parameter is introduced $C_3$, derived from a discrete second-order difference operator $D$ (Eq. (9)). $D$ represents a tridiagonal matrix where the main diagonal elements are equal to $-2$ and elements of the diagonals above and below equal to 1 (discrete representation of the second derivative), and $\epsilon$ is a regularisation parameter that determines the weight of this smoothness constraint relative to the other two terms.

The final mass emission rates are obtained by minimizing the total cost function $C$ using a standard optimization routine with the a-priori emission rates as the initial guess. This approach ensures that the calculated ash emission rates are consistent with both the observed data and the a-priori emissions estimates, while also favoring a smooth vertical distribution of emissions.

The inversion scheme presented in this study is not limited to Mt. Etna but can be applied to other volcanic eruptions worldwide, provided that suitable observational data are available. The methodology relies on ground-based lidar measurements, dispersion modeling (FLEXPART-WRF), and an inversion algorithm to estimate volcanic ash emissions. Therefore, it can be adapted to different volcanic settings where lidar observations or other remote sensing data like: (i) satellite-based lidars (CALIPSO, EarthCARE), (ii) geostationary satellites (SEVIRI), that are available to constrain the source term. Additionally, the approach can be extended to a regional or global scale by integrating multiple observation sites from lidar networks such as ACTRIS/EARLINET or incorporating additional satellite data. This would allow for improved ash emission estimates for various volcanic eruptions worldwide. Furthermore, the use of high-resolution wind field data (such as Aeolus or future wind lidar missions) can enhance the accuracy of dispersion forecasts in different geographic regions (possibly lacking sufficient information from radiosondes), making the methodology widely applicable for volcanic ash monitoring and forecasting.

### 3.4.1 A-priori source emissions $x^a$

To constrain the variability of the retrieved parameters and enhance the stability of the inversion outcome, a-priori emissions are also used inversion scheme. We determine the a-priori MER for ash particles following the approach outlined by Scollo et al., 2019 by inverting the observed plume heights over the Etna summit crater from VONA reports and field observations as observed by the INGV observatory, using the 1-D plume model (Degruyter and Bonadonna, 2012) as described in Sect. (3.3). Additionally, the London VAAC employs the same empirical relationship between observed plume heights and eruptive mass as proposed by Mastin et al., (2009), assuming a uniform vertical ash distribution.

The column heights of the ash plume from the 12 March 2021 were obtained from the ECV calibrated camera operated by the INGV-OE (Calvari et al., 2021; Corradini et al., 2018; Scollo et al., 2019) during the time period of 6:30 to 10:30 UTC (see Table A 1). The ash plume height reached up to 9.0 km a.s.l. In order to calculate the a-priori emissions, the data were resampled at ~2-hour intervals, specifically at 6:00 (from 6:30 to 7:45 UTC), 8:00 (from 8:00 to 9:45 UTC) and 10:00 UTC

(from 10:00 to 10:30 UTC). During the initial hours of the eruption (6:30 - 7:45 UTC) the ash plume was weak (Figure 1, a and Table A 1) with an average column injection height of 5.8 km, resulting in an estimated MER of approximately 12,000 kg/s according to the equation by Mastin et al., (2009) (Table 3). After 07:45 UTC, a stronger plume formed extending vertically above the vent (Figure 1b and Table A 1). The ash plume exceeded the ECV camera field of view (e.g., more than 9.0–9.5 km a.s.l.) and was particularly strong between 08:00 – 09:45 UTC (Figure 1c and Table A 1). The MER during this period averaged 58,800 kg/s, with a mean plume height of 10 km a.s.l. The standard deviation of the mean MER indicates considerable inconsistency in the emissions, as the MER can change rapidly during an eruption due to fluctuations in the eruptive dynamics, such as the collapse of the eruption column (Table 3). The ash plume height began to decrease several minutes after the lava fountain ceased (Figure 1d and Table A 1), with its disappearance becoming evident only after 10:15 UTC (Figure 1d). The MER during this phase (10:00 - 10:30 UTC) was approximately 6,300 kg/s (Table 3). The maximum plume elevation was not recorded by the ECV camera due to its limited field of view (approx. 9.0 – 9.5 km a.s.l. as noted by Scollo et al., 2014).However, according to SEVIRI aboard the geostationary Meteosat Second Generation satellite, the volcanic Ash Cloud Top Height (ACTH) between 08:15 to 08:45 UTC was estimated at 11.5 km a.s.l. (Calvari et al., 2021). This higher SEVIRI-derived plume height was used in the calculations for the a-priori ash emissions during this time window, as it provides a more accurate representation of the plume height at the peak of the eruption (see Appendix Table A 1).

Table 3: A-priori source vector $x^a$

| Time (UTC) | Mean Column Height (m) | Mean Mass Eruption Rate (MER) (kg/s) | Standard deviation (std) of Mean MER (kg/s) |
|---|---|---|---|
| 2021-03-12 06:00:00 | 5850 | 12,000 | 10,200 |
| 2021-03-12 08:00:00 | 10000 | 58,800 | 35,000 |
| 2021-03-12 10:00:00 | 5300 | 6,320 | 5,120 |

## 4. Results

On 12 March 2021 the Etna volcanic plume was captured over the PANGEA-NOA observatory by the Polly[XT] lidar system. A three-hour time window from 18:30 to 21:30 UTC, was selected to calculate aerosol optical properties using the Raman method (Ansmann et al., 2011b). This time-window was chosen based both on the lidar observations and on FLEXPART simulations, which also indicated the presence of ash particles over the PANGEA-NOA station. Figure 2 shows the time-height evolution of Polly[XT] lidar measurements, depicting a dense aerosol layer between 8 and 12 km a.g.l., with the majority of the ash plume (large, depolarizing aerosols) confined in the altitudes between 9 to 11 km approximately 11 hours after the eruption (18:30 – 21:30 UTC). The layer is associated with volcanic ash advection from Etna, as indicated by the high volume linear depolarization ratios (40 – 50 % at the center of the plume), which are typical of non-spherical volcanic ash particles (Gasteiger et al., 2011; Groß et al., 2013; Tackett et al., 2023; Wiegner et al., 2012), (Figure 2 b).

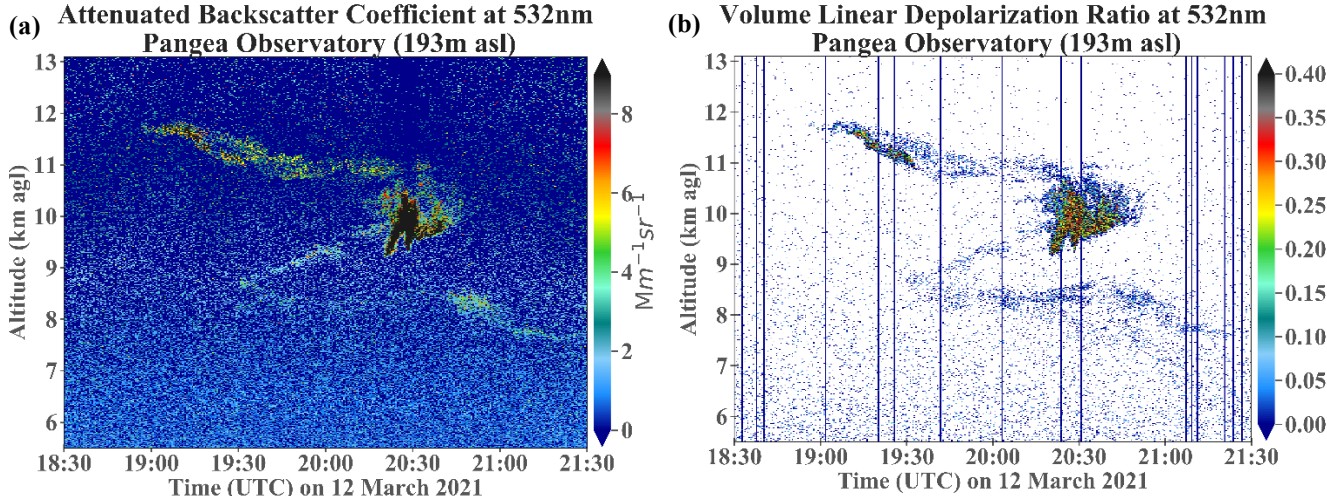

**Figure 2: (a) The time-height curtain plot of the attenuated backscatter coefficient and (b) the volume linear depolarization ratio at 532 nm based of Polly[XT] lidar observations at the PANGEA-NOA observatory during 12 March 2021 (18:30 to 21:30 UTC). Station elevation is at 193 m a.s.l. Altitude heights are given in kilometers above ground level. The blue vertical lines on (b) indicate negative values, which arise due to a low signal-to-noise ratio (SNR) of the measurements, and they are masked before data averaging and final retrievals.**

To further analyze the volcanic plume distribution, Figure 3 presents the mass eruption rates in kg/s for both the a-priori (represented by circles) and a-posteriori (represented by stars) values, plotted as a function of ash plume height (km a.s.l.) and eruption time (UTC). The a-posteriori ash particle emissions in the "w" Aeolus simulation (Figure 3 a, b), obtained through the inversion scheme presented herein, were used as input for a new FLEXPART forward run. As discussed in Sect. 3.4.1, the a-priori MER (Figure 3 a, b) for ash particles was determined using the approach outlined by Scollo et al. (2019). The a-priori MER was obtained by inverting observed plume heights from the VONA reports, based on data collected by calibrated cameras operated by the INGV-EO observatory. The ash plume's disappearance becomes noticeable only after 10:15 UTC (Figure 1d). The a-priori MER values represented by circles, exhibit significant variability throughout the eruption period on 12 March 2021, between 06:30 and 10:30 UTC. Peak MER values, approaching 80,000 kg/s, are observed at approximately 12 km altitude between 09:30 and 09:45 UTC. Additionally, notable peaks occur at lower altitudes between 08:15 and 09:00 UTC, where MER values reach approximately 45,000 kg/s at around 9 km altitude (Figure 3 a, b).

In contrast, the a-posteriori MER values, denoted by stars, display a more constrained and consistent pattern, with lower magnitudes across most altitudes and times with respect to the a-priori estimates. The maximum a-posteriori MER reaches approximately 45,000 kg/s at an altitude of 10.5 km, occurring between 08:15 and 08:45 UTC (Figure 3 a, b).

The eruption dynamics involve a complex evolution of the volcanic plume, with phases of rising and collapsing. However, this dynamic behavior is not explicitly resolved in the a-posteriori simulations, which do not capture rapid fluctuations in plume height and intensity. Instead, the inversion algorithm adjusts the a-posteriori MER at each altitude over time,

dynamically increasing or decreasing emission rates to achieve the best agreement with available observations. The most significant refinement occurs between 08:15 - 08:45 UTC, within the 8 - 12 km altitude range, where lidar observations provide direct constraints on the plume's vertical structure. As a result, the inversion optimizes the emission estimates primarily within this altitude range, ensuring the highest degree of agreement between observed and a-posteriori emissions.

A notable distinction between the two sets of emission estimates is the greater spread of the a-priori emissions across a wider

range of altitudes, with values often exceeding those of the a-posteriori emissions. This is especially evident at lower altitudes (below 7 km) (Figure 3 b, c), where relative differences range between 40 % and 80 % from 06:30 to 07:45 UTC. These differences suggest an overestimation of the initial a-priori emissions obtained by inverting observed plume heights from the VONA reports, compared to the ash emissions derived from the inversion scheme (Figure 3 c).

On the other hand, the a-posteriori MER values present a more refined and clustered distribution between 8 and 12 km altitude

(Figure 3 a), indicating a more constrained and likely more accurate estimation of ash emissions. This contrast is particularly evident when compared to the more scattered and variable a-priori estimates.

Between 10:00 and 10:30 UTC both a-priori and a-posteriori estimates indicate a distinct decline in MER, with values dropping below 10,000 kg/s at lower altitudes (~5 km). During this period, the relative differences between plume height and MER exceed 80 %, highlighting the divergence between the initial and adjusted estimates (Figure 3 c).

Regarding the ash plume height, the a-posteriori estimates consistently indicate higher altitudes compared to the a-priori estimates, a discrepancy potentially attributed to the limited field of view of the calibrated camera from the INGV-EO observatory. The camera's restricted range (approximately 9.0 – 9.5 km a.s.l., as noted by (Scollo et al., 2014) may have failed to capture the full extent of the plume, leading to underestimations in the a-priori estimates.

Additionally, Calvari et al. (2021) further indicate that the observed plume column altitudes predominantly range between 6

and 9 km, which is the upper limit of the INGV-OE camera system. As a result, column heights exceeding 9 km a.s.l. are likely limited, contributing to differences between a-priori and a-posteriori estimates.

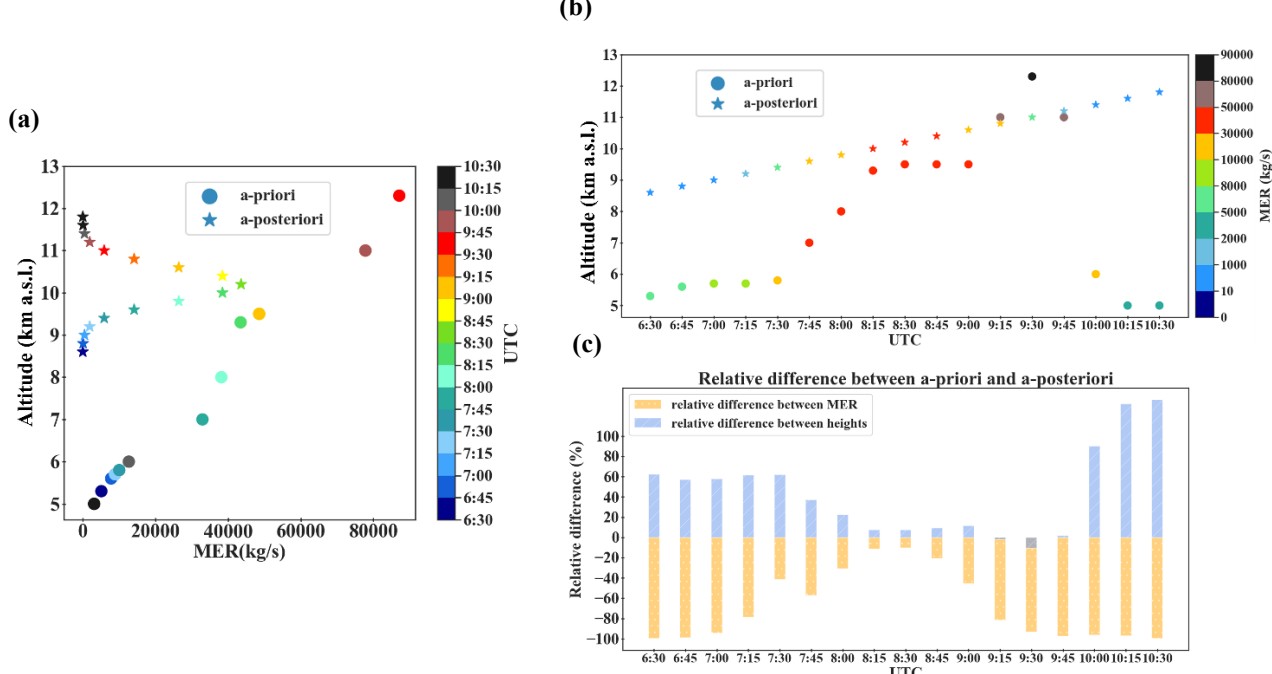

**Figure 3: A-priori and a-posteriori ash emissions. (a) Comparison of temporally averaged vertical profiles of ash emissions used a priori (circles) and obtained a posteriori by the inversion (stars). The colorbar indicates the corresponding times of eruption, (on 12 March 2021, from 6:30-10:30 UTC), each color representing a specific time; (b) A-priori (circles) and a-posteriori (stars) MER (unit kg/s) as a function of altitude (km) and time (UTC), on 12 March 2021 from 6:30 - 10:30 UTC. The colorbar indicates the corresponding MER values, (from 0 to 90,000 kg/s), each color representing a specific MER range. The time axis reflects the period during which the ash plume was recorded by the ECV calibrated camera (6:30 - 10:30 UTC); (c) Relative differences (%) between a-posteriori and a-priori for ash emissions (orange columns) and plume height (blue columns) as a function of time (UTC). All heights are given in kilometers above sea level.**

The relative differences between the two estimates are notably smaller, ranging from 10 % to 40 % between 08:00 and 09:00 UTC (Figure 3 c), suggesting a reasonable agreement between the a-priori and a-posteriori assessments for both emissions and column heights during this time window of the eruptive phase.

This improvement in the a-posteriori profile underscores the efficacy of the inversion algorithm in producing a more reliable representation of the vertical distribution of the ash emissions by improving the precision of eruption source parameters. The a-posteriori MER profile alignment with the observational data suggests that this method provides a robust and realistic assessment of ash emissions, particularly in the critical altitude range where volcanic plumes typically occur (Degruyter and Bonadonna, 2012; Mastin et al., 2009; Scollo et al., 2019).

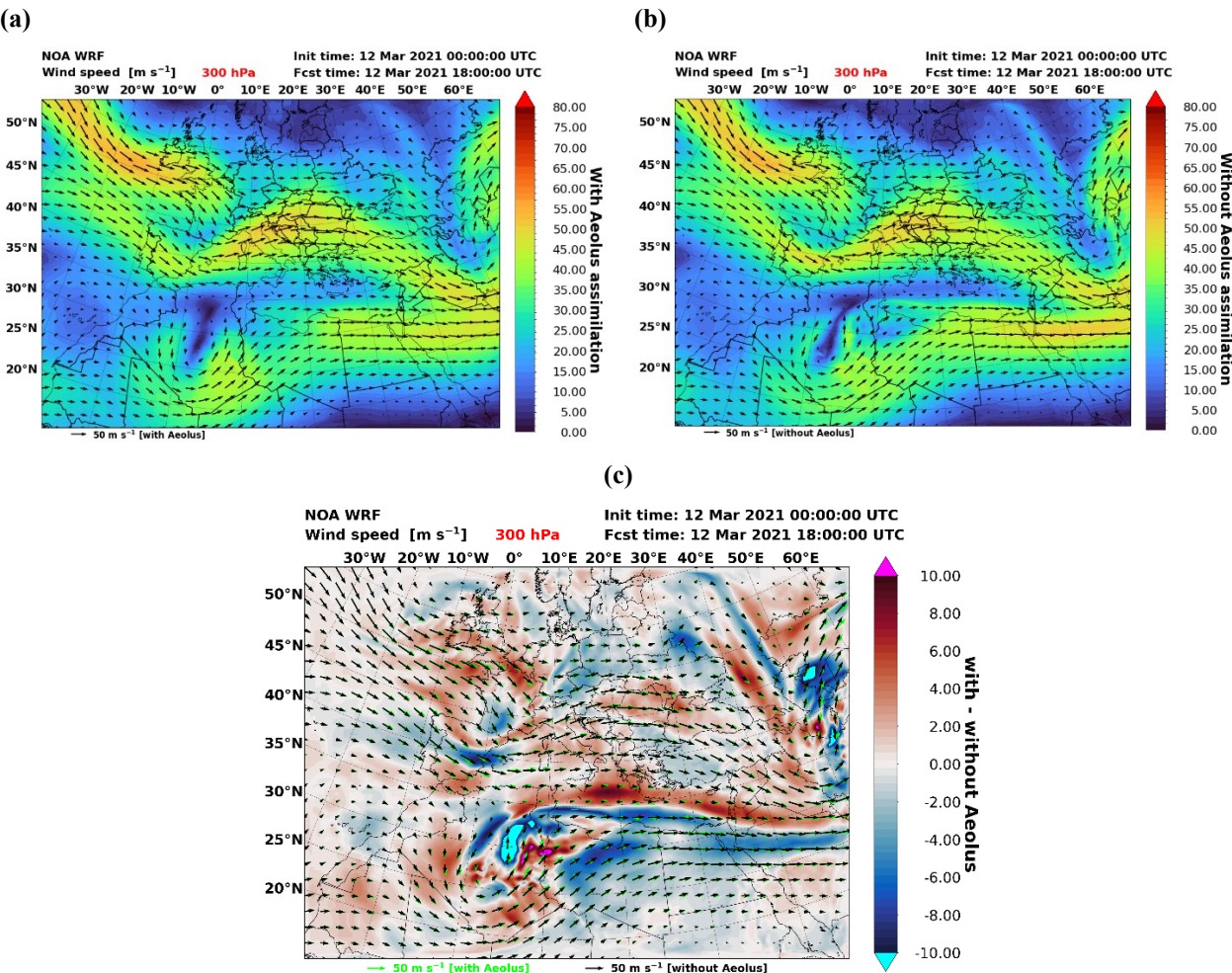

**Figure 4: Wind speed (m/s) in WRF 18 h forecasts. Horizontal winds (a) "w" Aeolus assimilation, (b) "w/o" Aeolus assimilation and (c) wind speed differences ("w" – "w/o" Aeolus assimilation) at 300 hPa (~9.6 km).**

The transport and dispersion of volcanic ash particles are strongly influenced by upper-air circulation patterns, which play a crucial role in determining the trajectory and lifetime of the volcanic plume. To assess the sensitivity of the volcanic ash transport to the driving meteorology, two simulations were performed using the WRF regional model over the study period. These simulations were driven by two versions of the ECMWF-IFS global model: one incorporating Aeolus wind profile assimilation ("w") and one without Aeolus assimilation ("w/o") (see Sect. 3.3). To evaluate the influence of upper-level circulation on volcanic ash transport, wind maps at 100, 200, 300, and 500 hPa, were generated, for the period of significant volcanic activity (Figure A 3-Figure A 5).

Given that lidar observations estimated the volcanic plume's center of mass at approximately 10 km, the analysis primarily focused on the 300 hPa level (~9.6 km), which closely corresponds to this altitude (Figure 4). Analyzing the WRF regional model wind vectors at upper-tropospheric levels (300 hPa, ~9.6 km) at 18:00 UTC (approximately 11 hours after the Etna

eruption), the general atmospheric circulation remained predominantly zonal over the Mediterranean, with westerly winds prevailing throughout the troposphere. Over the Anatolian Plateau and the Eastern Mediterranean Sea, these winds transition into northwesterlies, favoring the direct transport of the Etna plume towards Greece and the Eastern Mediterranean.

A comparison of the two simulations ("w" and "w/o" Aeolus assimilation) indicates that the overall atmospheric pattern remains consistent, with the subtropical and polar jet streams dominating the circulation. However, notable differences in wind speed are evident, as highlighted in the wind speed difference map for the WRF 18-hour forecast (Figure 4).

The color shading in Figure 4 (c) illustrates the differences between the two WRF runs on 12 March 2021 (18:00 UTC). This comparison indicates significant strengthening of winds at 300 hPa when Aeolus wind profiles are assimilated (Figure 4 c), with maximum difference values reaching approximately 8 m/s. Additionally, slight differences in wind vector direction ("w" Aeolus (green) and "w/o" Aeolus (black)) are observed, particularly over the Ionian Sea (from W to NW) and the Eastern Mediterranean between Crete and Cyprus (from WNW to NW), where the two jet streams merge.

Similar wind speed tendencies are observed at 200 hPa (Figure A 4). In contrast, at 500 hPa (Figure A 5), the influence of Aeolus assimilation is less pronounced, indicating that the most significant differences occur at higher altitudes where jet stream dynamics dominate.

At 100 hPa (Figure A 3), a strong westerly jet stream is evident across Europe and North Africa, indicating fast-moving winds that could contribute to long-range transport of volcanic particles. The corresponding wind speed difference map (Figure A 3 c) shows high differences mainly along the jet stream axis, suggesting that Aeolus assimilation plays a crucial role in improving the representation of high-altitude wind fields critical for long-range ash transport.

These findings highlight the importance of accurate upper-air circulation representation in volcanic ash transport modeling. The inclusion of Aeolus wind profiles in the ECMWF-IFS model leads to a more refined depiction of wind patterns, particularly at upper tropospheric and lower stratospheric levels, which are crucial for accurately forecasting the dispersion of volcanic emissions.

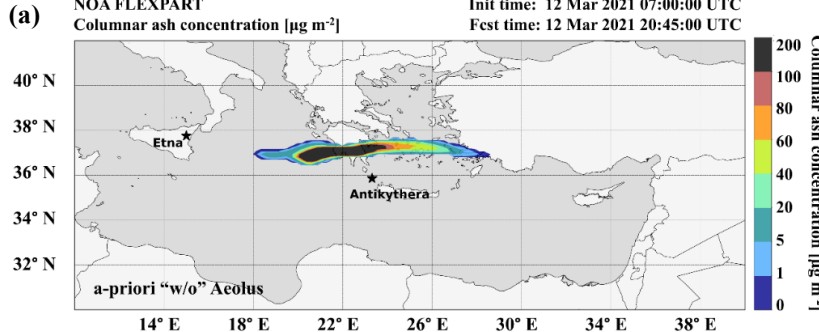

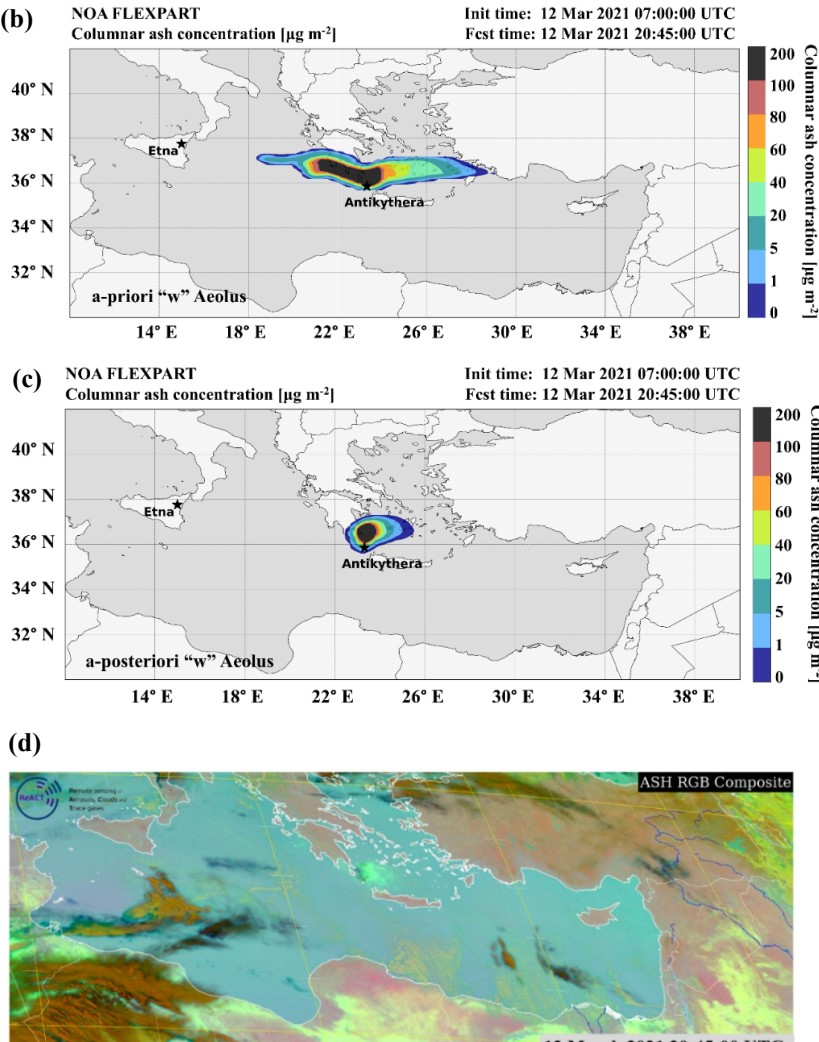

**Figure 5: FLEXPART simulations of the volcanic ash. (a) a-priori ash column loading (µg/m²), using meteorological fields "w/o", and (b) "w" Aeolus wind assimilation; (c) a-posteriori ash column loading (µg/m²), using meteorological fields "w", (12 March 2021, 20:45 UTC); (d) EUMETSAT Meteosat-11 volcanic Ash (RGB-MSG-0-degree) product of the ash plume derived from the Spinning Enhanced Visible and InfraRed Imager (SEVIRI), during paroxysmal activity at Mt. Etna on 12 March 2021. Composite thermal IR (8.7, 10.8, 12 wavelengths) satellite image from the SEVIRI captures the volcanic ash plume about 11 hours after the start of the eruption above the PANGEA-NOA station, at Antikythera island in Greece, on 12 March 2021, 20:45 UTC. SEVIRI data can downloaded from the EUMETSAT data portal (https://view.eumetsat.int/productviewer?v=default).**

The FLEXPART simulated a-priori distribution (µg/m²) of the ash clouds over the Eastern Mediterranean at 20:45 UTC, using meteorological fields "w/o" and "w" Aeolus wind assimilation is shown in Figure 5 (a, b). The ash plume is shown to arrive over Antikythera from the west, only when Aeolus assimilated wind fields were used (Figure 5 b). In contrast, the volcanic plume in the "w/o" Aeolus forecast never crosses Antikythera, as the forecasted cloud displaced to the north (Figure 5 a).

Additionally, the a-posteriori distribution of the ash plume transport (μg/m²) over the Eastern Mediterranean at 20:45 UTC, using Aeolus wind assimilation, is shown in Figure 5 (c). However, the a-posteriori particle emission rates in the "w/o" Aeolus simulation, could not be estimated from the inversion scheme due to very low source-receptor relationships derived from the FLEXPART model (see Appendix A, Figure A 2 right panel). As a result, the a-posteriori simulation of ash plume transport "w/o" Aeolus assimilation was not produced.

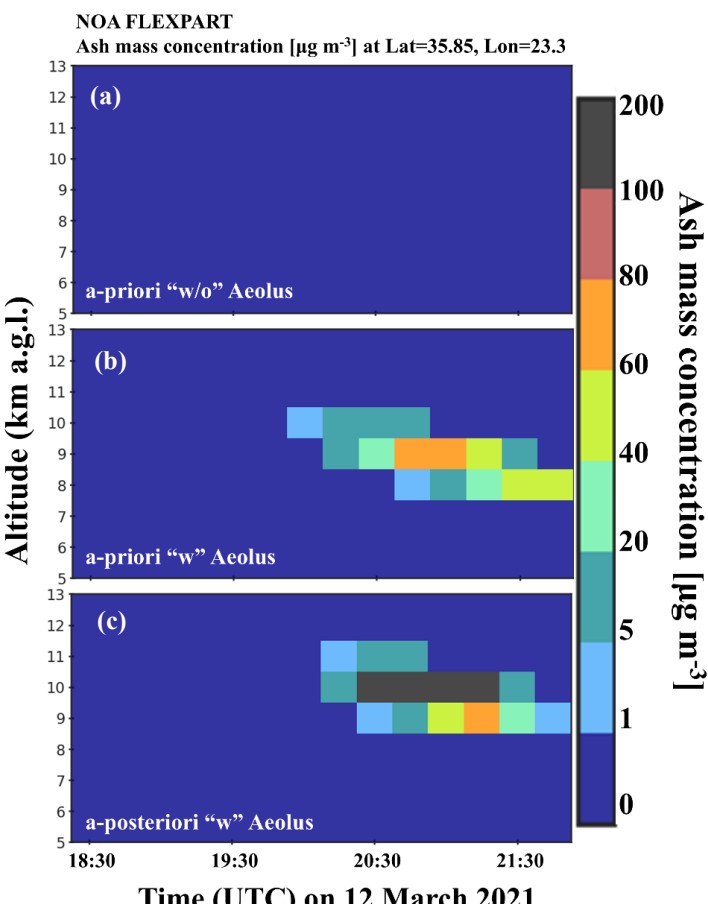

**Figure 6: FLEXPART time-height cross-sections on 12 March 2021, 18:30 – 21:30 UTC, over the PANGEA observatory in Antikythera, Greece. (a) time-height plot of a-priori FLEXPART volcanic ash concentrations (μg/m³), over Antikythera "w/o" Aeolus wind assimilation over Antikythera, Greece (zero values); (b) time-height plot of a-priori FLEXPART volcanic ash concentrations (μg/m³) over Antikythera "w" Aeolus wind assimilation; (c) time-height plot of a-posteriori FLEXPART volcanic ash concentrations (μg/m³) over Antikythera "w" Aeolus wind assimilation over Antikythera, Greece ("w/o" are not calculated). All heights are given in kilometers above ground level.**

The a-posteriori ash plume is notably more concentrated than the a-priori plume (Figure 5 b and c) and covers a smaller area, mostly limited to the area around Antikythera and southern Greece. In contrast, the a-priori ash plume (Figure 5 b) is more

widely dispersed potentially due to the higher MER values (Figure 3 a denoted as circles) leading to an overestimation of the

a-priori ash emissions. The a-priori ash plume dispersion extends from the eastern coast of Greece and reaches as far as the western islands. Furthermore, the structure of the a-posteriori ash plume closely resembles the ash cloud image captured by the EUMETSAT Meteosat-11 Ash RGB product from the SEVIRI satellite, above Antikythera island on 12 March 2021, at 20:45 UTC (Figure 5 d), again highlighting the importance of constraining the variability of the simulation results towards a more stable solution.

A thorough evaluation of the different model simulations is performed against the quality-assured lidar measurements of the PANGEA-NOA observatory. Figure 6 represents the vertical profiles of the FLEXPART simulated ash mass concentrations over PANGEA-NOA. FLEXPART vertical time-height cross-sections of volcanic ash a-priori and a-posteriori concentrations "w" Aeolus assimilated fields (Figure 6 b, c) show a similar pattern to the observed volcanic aerosol layer over Antikythera (Figure 2 a) but reveal significant differences in the vertical distribution and ash mass concentrations.

Specifically, the a-priori simulation using "w" Aeolus wind assimilation, forecasts a volcanic ash layer at an altitude range of approximately 7.5 to 11 km a.s.l., with ash concentrations reaching bellow to 100 µg/m³ over Antikythera between 18:30 UTC and 21:30 UTC (Figure 6 b). In contrast, the a-priori run "w/o" Aeolus assimilation fails to capture the observed ash particle concentrations over Antikythera (Figure 6 a). In the a-posteriori simulation, the ash plume driven by the Aeolus wind fields is notably more aligned and better defined than in the a-priori simulation with respect to the observed ash plume (Figure 6 c) and

(Figure 2). The a-posteriori profile reveals a volcanic ash layer at an altitude range of 8 to 12 km with higher ash concentrations, than in the a-priori layer, reaching up to 200 µg/m³ over Antikythera during the same time period (Figure 6 c). Notably, in the a-posteriori profile (Figure 6 c), the main part of the ash plume with the highest concentrations is confined between 9 and 11 km, consistent with the observed lidar profile (Figure 2a). However, the a-posteriori FLEXPART time-height cross-sections using the "w/o" Aeolus wind fields were not calculated, as the a-posteriori emission rates could not be estimated by the

inversion scheme due to very low SRR derived from the FLEXPART model (see Appendix A, Figure A 2 right panel).

The better agreement in both the vertical distribution and the concentration of the volcanic ash in the a-posteriori simulation (Figure 6 c), compared to the time-height profile of the observed ash plume derived from Polly$^{XT}$ lidar, on 12th of March from 18:30 to 21:30 UTC (Figure 2 a), highlights the effectiveness of the inversion process when utilizing Aeolus wind data.

The aerosol optical properties profiles retrieved from the lidar data, are shown in Figure 7. The POLIPHON method as

described in Sect. 3.1.1. was utilized to derive the pure-ash mass concentration profiles.

Polly$^{XT}$ lidar retrievals show that the volcanic ash concentrations over PANGEA-NOA reached up to almost $250 \pm 80$ µg/m³ at the plume's center of mass which is estimated at 10 km a.g.l. (orange line at Figure 7 c and Figure 7 d). The uncertainty in mass concentration calculation is marked with a black error bar in Figure 7 a.

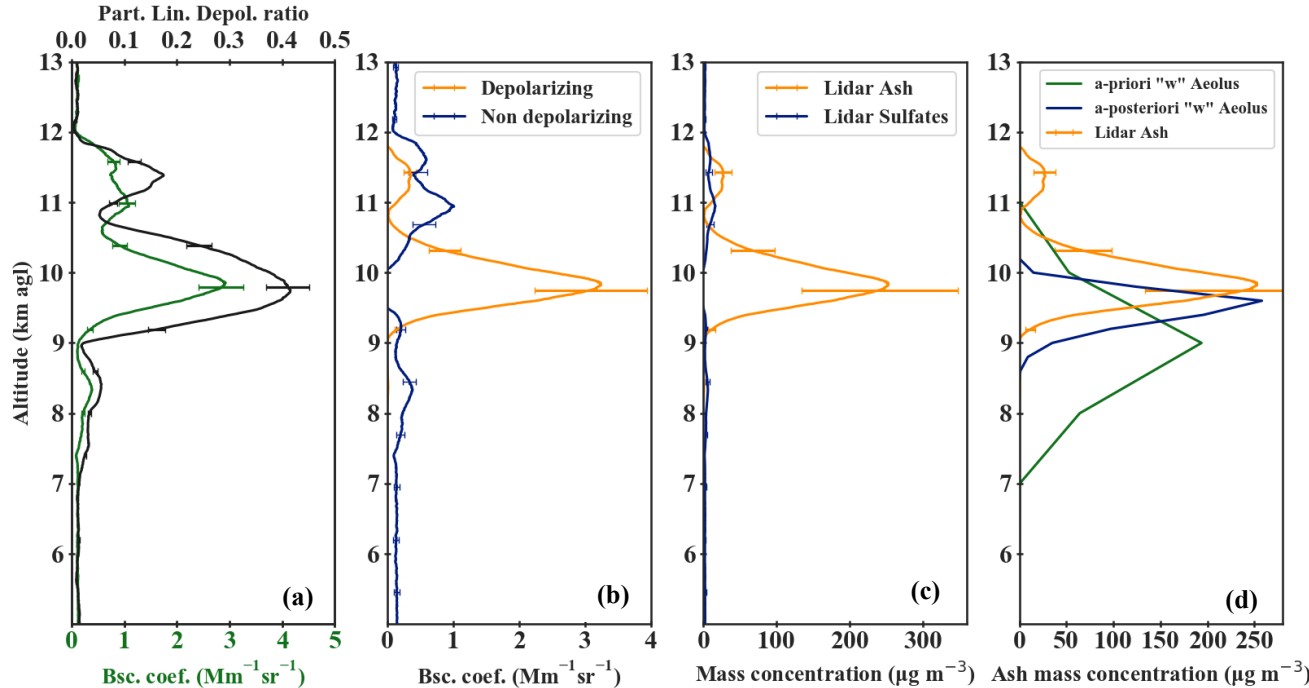

Figure 7: Lidar-derived optical properties over the PANGEA observatory on 12 March 2021 (18:30 – 21:30 UTC). Vertical distributions of: (a) total backscatter coefficient (green line) and particle linear depolarization ratio at 532 nm (black line); (b) depolarizing (orange line) and non-depolarizing (blue line) particle backscatter coefficient; (c) Volcanic mass concentrations using the POLIPHON method for ash (orange line) and sulfates (blue line); (d) Vertical profile of volcanic ash. Volcanic ash mass concentrations using the POLIPHON method (orange line); FLEXPART a-priori model simulations "w" Aeolus assimilated winds (green line) result of Amiridis et al., (2023), for the fine particles (3, 5, 9, and 21 μm diameter); FLEXPART a-posteriori model simulations "w" Aeolus assimilated winds (blue line), for the fine particles (3, 5, 9, and 21 μm diameter); a-priori and a-posteriori ash mass concentrations "w/o" Aeolus simulation equals to zero and are not shown. All heights are given in kilometers above ground level.

The a-posteriori ash emissions from the "w" Aeolus simulation, obtained through the inversion scheme (Figure 3, stars), were used as input for a new FLEXPART forward run. This run was conducted to estimate a-posteriori ash mass concentrations above the PANGEA-NOA station between 18:30 and 21:30 UTC, focusing on fine ash particles with diameter of 3, 5, 9, and 21 μm.

Figure 7 d compares the vertical profiles of the observed and the simulated (a-priori and a-posteriori "w" Aeolus assimilation) volcanic ash concentrations. The a-priori and a-posteriori ash mass concentrations "w/o" Aeolus simulation equals to zero and are not shown, as the SRM derived using 'w/o' Aeolus assimilated wind fields results in negligible sensitivities. The volcanic ash plume in the 'w/o' Aeolus simulation never reached Antikythera on 12 March 2021 between 18:00-21:30 UTC due to a northward shift (Figure 5 a).

The corresponding mass concentrations derived from FLEXPART a-priori simulation (green line) and a-posteriori simulation (blue line) are shown in Figure 7 d for comparison with the lidar observations (orange line).

The a-priori simulation produced ash concentrations of approximately 150 - 180 μg/m³ at the plume's center of mass, at 8.5 km a.g.l. (green line in Figure 7 d). While the a-priori profile shows good spatio-temporal agreement with the lidar retrievals (orange line in Figure 7 d), there is a slight vertical shift of 1 km between the modeled and observed ash mass peaks, which is critical for aviation safety. Furthermore, there is a misfit of about 50 μg/m³ between the ash concentrations derived by the PollyXT lidar and those reproduced by the model in the a-priori simulation, even with Aeolus data assimilated.

In contrast, when comparing the modeled a-posteriori ash mass concentrations to the lidar observations their agreement is evident when Aeolus winds are assimilated. The maximum ash mass concentration is approximately 250 μg/m³ at 9.8 km, closely matching the peak observed by the lidar, while also the vertical distribution of the ash plume is depicted with high accuracy. The difference between the observed and the a-posteriori simulated ash mass concentrations is minimal and only 2 %. In contrast, the difference between the lidar observations and the a-priori ash simulations ranged from 28% to 40%. This demonstrates that the estimated emission profile obtained from the inversion algorithm that presented herein is remarkably robust. Overall, the inversion profile yields a much better agreement with lidar observations, confirming the effectiveness of the inversion process and the value of incorporating Aeolus wind data into the model.

## 5. Conclusions and discussion

The present study presented an inversion method to estimate the volcanic emission rate profile with a Lagrangian particle dispersion model and a ground-based lidar system. The technique was applied to the case study of the explosive eruption of Mt. Etna, Italy, on 12 March 2021. To assess the impact of Aeolus wind assimilation in volcanic ash dispersion forecasts, the simulation was repeated twice: once with Aeolus data assimilated ("w" experiment) and once without ("w/o"). The volcanic aerosol layers observed above the PANGEA-NOA station in Antikythera, along with the clear sky conditions in the days after the eruption, made this an ideal test case. Important conclusions from our work are as follows:

The PollyXT lidar system of PANGEA-NOA detected a dense aerosol layer between 8 and 12 km, with the volcanic ash plume primarily concentrated between 9 and 11 km. FLEXPART simulations, both a-priori (with an empirical emission profile) and a-posteriori (with the emission profile produced by the inversion algorithm), were conducted to derive the modeled plumes vertical distribution and concentration. The a-priori "w" Aeolus simulation showed a broader dispersion of the ash plume potentially due to the overestimation of the a-priori ash emissions obtained by inverting observed plume heights from the VONA reports, whereas the a-posteriori simulation, based on the inversion results, produced a more refined and consistent ash plume profile, confined to a smaller area, mostly around Antikythera and southern Greece, closely similar to the ash cloud observed by the SEVIRI satellite.

In terms of ash mass concentration, the a-priori profile with Aeolus wind data assimilated, shows a good spatio-temporal agreement with the lidar retrievals but exhibited a slight vertical shift of 1 km with respect to the observed ash mass peaks (Amiridis et al., 2023) along with a misfit in mass concentrations of about 50 μg/m³, a critical factor for aviation safety. In contrast, the a-posteriori ash mass concentrations demonstrate a better agreement with the observations above PANGEA when

Aeolus winds are assimilated. The maximum ash mass concentration is found close to 255 µg/m³ at 9.8 km, closely matching the peak observed by the lidar, depicting a minimal difference of the order of 2 % between the observed and the a-posteriori

simulated ash mass concentrations. In contrast, the difference between the lidar observations and the a-priori ash simulations ranged from 28% to 40%. This consistency highlights the robustness of the new inversion algorithm and the significant improvement in the vertical distribution and the ash mass concentration. However, additional independent datasets, such as ground-based, satellite remote sensing data, or airborne in-situ measurements along the plume's trajectory, would further enhance the validation of this methodology and should be considered in future studies.

To further assess the reliability of the retrieved emissions, a Monte Carlo error propagation analysis was conducted, introducing normally distributed perturbations to the lidar measurements. With this method the standard deviation of the retrieved emissions at each height level was estimated. The results indicate that the inversion output remained highly stable, with minimal variation across Monte Carlo realizations, suggesting that the single-station observational setup does not introduce significant uncertainty. To enhance the sensitivity of the inversion framework and provide a more comprehensive uncertainty

assessment, multiple lidar stations or complementary remote sensing techniques are essential.

The accuracy of the FLEXPART a-posteriori simulation is highly dependent on the precision of the driving meteorological fields ("w" Aeolus wind fields), as well as on volcano source parameters such as the plume height and the mass eruption rates, which are refined through the inversion process (a-posteriori MER).

The advantages of Aeolus wind assimilation for global NWP models have been well documented, particularly by Rennie et al.

(2021), who demonstrated significant improvements in wind field representation, especially in the Tropics and Southern Hemisphere. Further enhancements in wind forecasts were observed in the study of Amiridis et al. (2023), where regional NWP models benefited from Aeolus wind assimilation. Our case study validates these findings, showing that the assimilation of Aeolus wind profiles leads to a significant improvement in the estimation of volcanic emission rates in the vertical distribution, optimizing the agreement between lidar observations and a-posteriori model simulation.

Real-time applications, such as those of VAACs, demand a rapid response to volcanic ash hazards. Once the plume is detected and initial data from lidar systems become available, the presented method can quickly provide the necessary information to calculate the current and future position and extent of the plume within a few hours. This underscores the imperative for high-quality, rapidly accessible data, such as that provided by organized ground-based lidar networks employing standardized algorithms and procedures, such as those used by EARLINET a key component of the ACTRIS infrastructure.

However, their applicability to the proposed methodology depends on the operation of a backscatter-depolarization lidar, which constitutes the primary requirement. In cases where direct measurements of essential parameters, such as lidar ratios, are unavailable, values from the scientific literature can be used. A more advanced configuration, incorporating Raman lidar capabilities, would enhance the accuracy of retrieved backscatter and lidar ratio coefficients. Additionally, for daytime measurements, a co-located sun photometer would facilitate direct estimation of the conversion factors required in the

inversion process. Beyond ground-based applications, the methodology is also applicable to spaceborne aerosol lidars, which

provide vertical profiles of the backscatter coefficient and particle linear depolarization ratio, both fundamental parameters for the inversion process.

Furthermore, the methodology presented herein can be applied to current or future satellite missions that employ lidar measurements (e.g., the EarthCARE mission). While passive satellites offer near-global coverage of ash cloud measurements within minutes to hours, ground-based or satellite lidar systems provide more accurate direct retrievals of the vertical distribution within the ash plume column.

Our methodology is broadly applicable and efficient enough for real-time implementation. It can supply ash forecasting models with an objectively derived quantitative source term, leading to improved forecasts that are critical for the aviation sector. These enhanced forecasts provide more effective emergency responses, ensuring safer and more efficient flight operations during volcanic eruptions, while at the same time minimizing the risk of accidents and the financial impact of flight cancellations.

## Appendix A

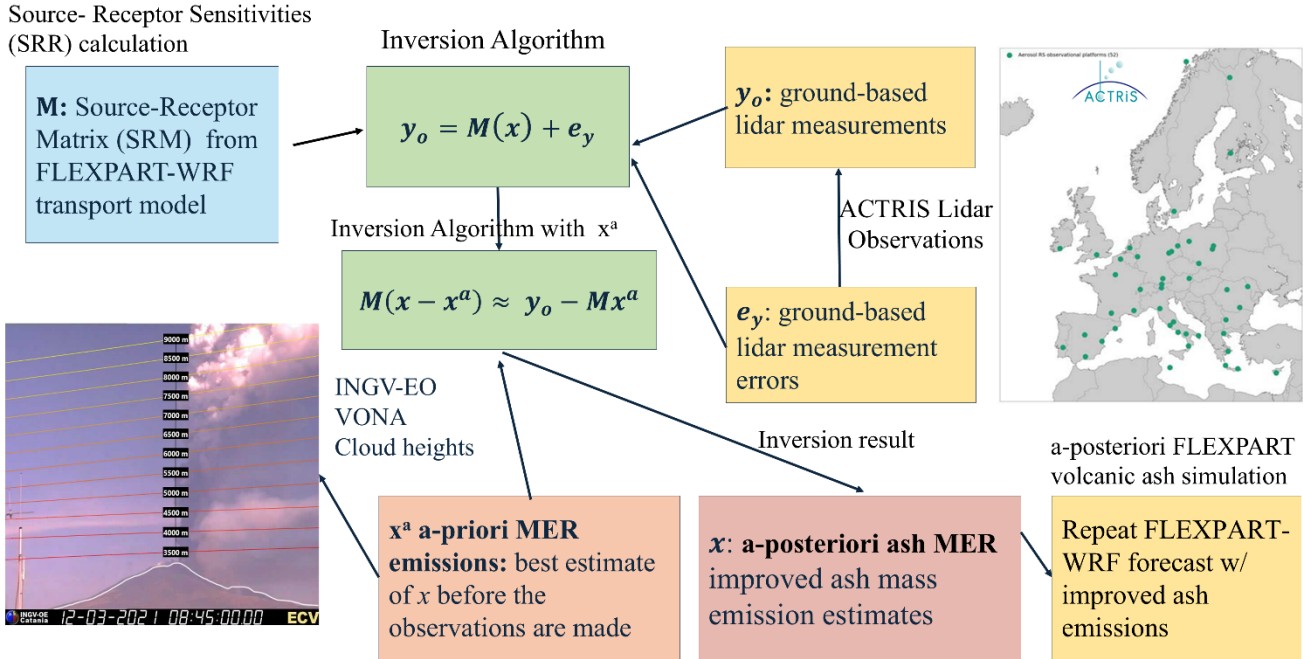

**Figure A 1: Workflow of the methodology.**

The SRR for a size distribution of volcanic ash particles with four size bins (3, 5, 9, and 21 μm diameter), derived from the FLEXPART model, using the "w" Aeolus assimilated wind fields, indicate that the volcanic emissions observed above the PANGEA-NOA observatory (receptor - y axis) on 12 March 2021 (from 18:30 to 21:30 UTC) at the height range 6 - 12 km

mostly originate from release heights between 5 and 11.5 km above the Etna volcano (source – x axis) (Figure A 2, left). These source release heights are consistent with the observed emissions above the PANGEA-NOA station, particularly when the particle release time was 06:00 - 08:00 and 08:00 - 10:00 UTC. The source heights for the fine particles align well with the eruptive column heights, as reported from the INGV-OE calibrated cameras (Figure 1). Additionally, the inversion algorithm was utilized with the FLEXPART SRR only for these two release times. In contrast, the SRR using "w/o" Aeolus assimilated wind fields show that the volcanic particles arriving above the PANGEA station at heights of 8 – 10 km (receptor y axis) are few and originate from release heights around 8 - 11 km above Etna (source x axis) and only when the particle release time was 04:00 - 06:00 UTC (Figure A 2, right). This release time is not accurate, as the eruption actually began at 06:00 UTC according to the VONA messages from INGV-EO.

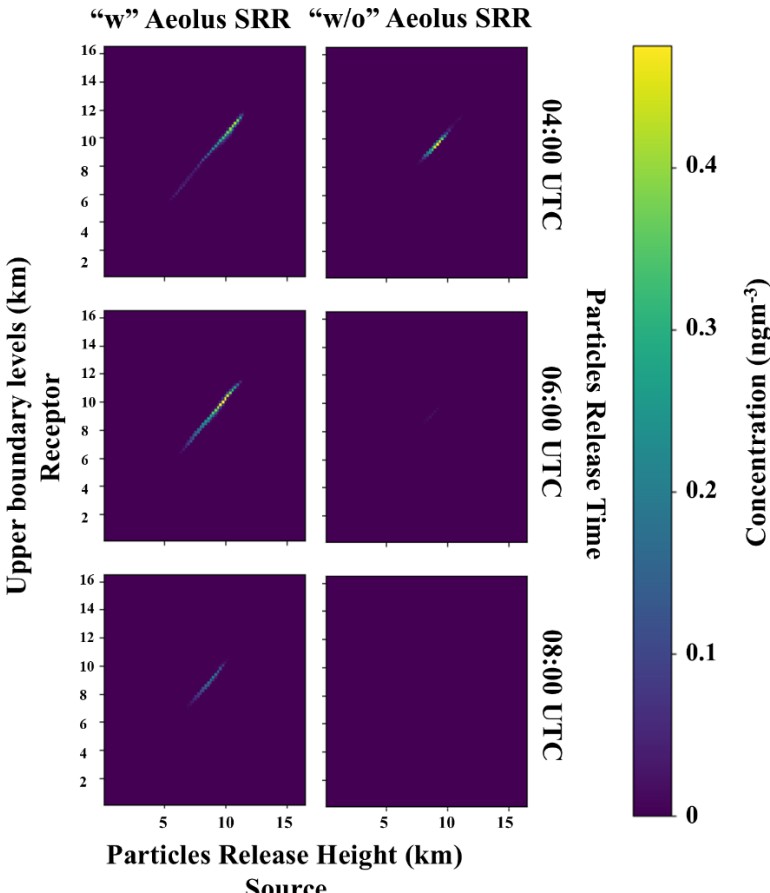

Figure A 2: Source-Receptor sensitivities for the fine particles (3, 5, 9, and 21 μm diameter) "w" Aeolus assimilated winds (left panel) and "w/o" Aeolus simulation (right panel). The horizontal axis "x" depicts the particles release height (km) above Etna and the vertical axis "y" is the altitude above PANGEA that the emissions observed on 12 March 2021 (18:30 to 21:30 UTC).

**Table A 1: Volcanic ash plume heights (m) during the eruption activity from 06:30 to 10:30 UTC, as recorded by the ECV camera operated by INGV-EO (second column) and adjusted heights incorporating SEVIRI satellite observations where applicable (third column).**

| TIME (UTC) | HEIGHT (m) from ECV camera (INGV-EO) | HEIGHT (m) incorporating SEVIRI observations |
|---|---|---|
| 6:30 | 4000 | 4000 |
| 6:45 | 5500 | 5500 |
| 7:00 | 5500 | 5500 |
| 7:15 | 6000 | 6000 |
| 7:30 | 6500 | 6500 |
| 7:45 | 7000 | 7000 |
| 8:00 | 7500 | 7500 |
| 8:15 | >9000 | 11500 |
| 8:30 | >9000 | 11500 |
| 8:45 | >9000 | 11500 |
| 9:00 | >9000 | 10000 |
| 9:15 | >9000 | 9500 |
| 9:30 | >9000 | 9500 |
| 9:45 | 9000 | 9000 |
| 10:00 | 6500 | 6500 |
| 10:15 | 5000 | 5000 |
| 10:30 | 4500 | 4500 |

700

(a)  (b)

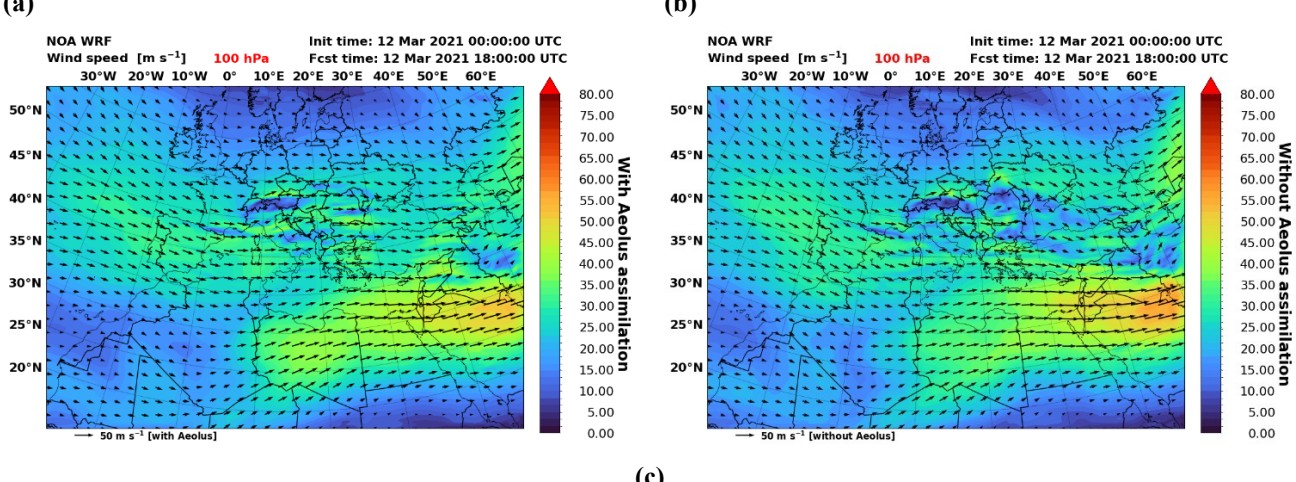

(c)

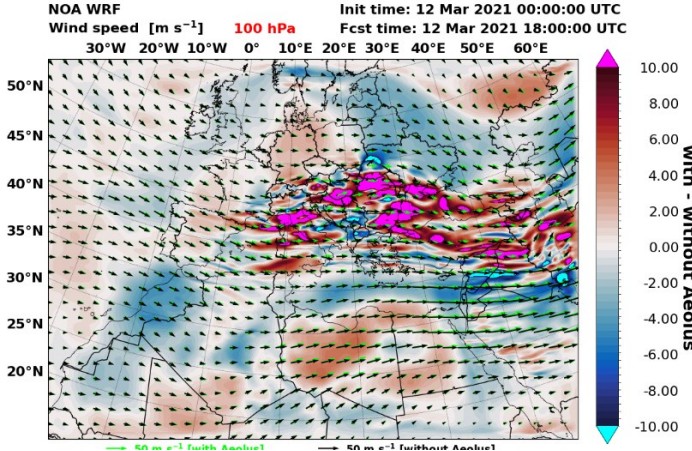

**Figure A 3: Wind speed (m/s) in WRF 18 h forecasts. Horizontal winds (a) "w" Aeolus assimilation, (b) "w/o" Aeolus assimilation and (c) wind speed differences ("w" – "w/o" Aeolus assimilation) at 100 hPa (~ 16 km).**

**(a)**                                                                                      **(b)**

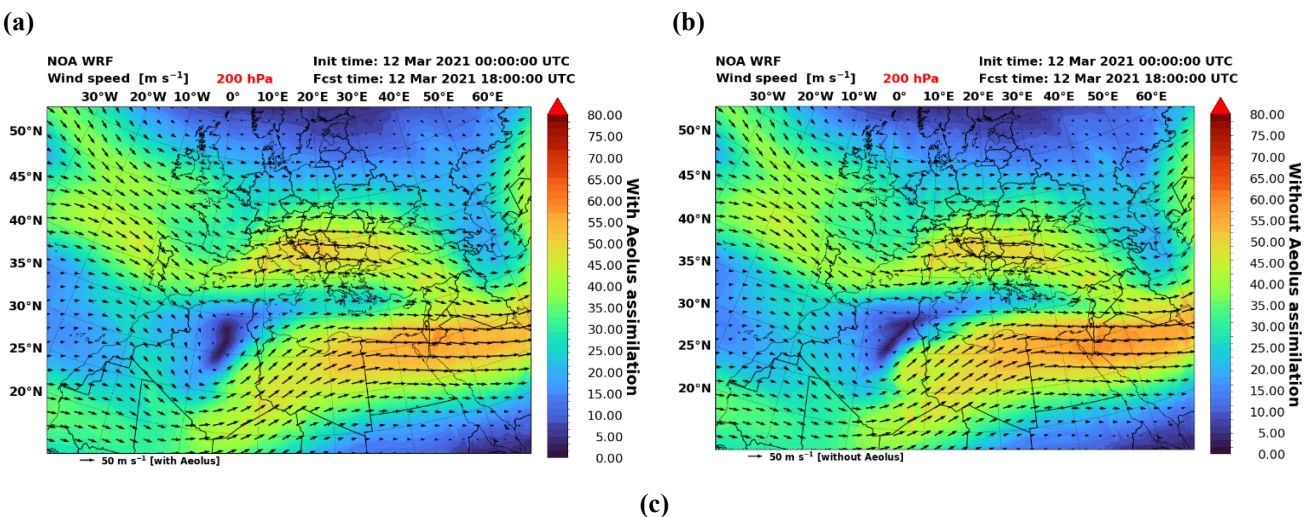

**(c)**

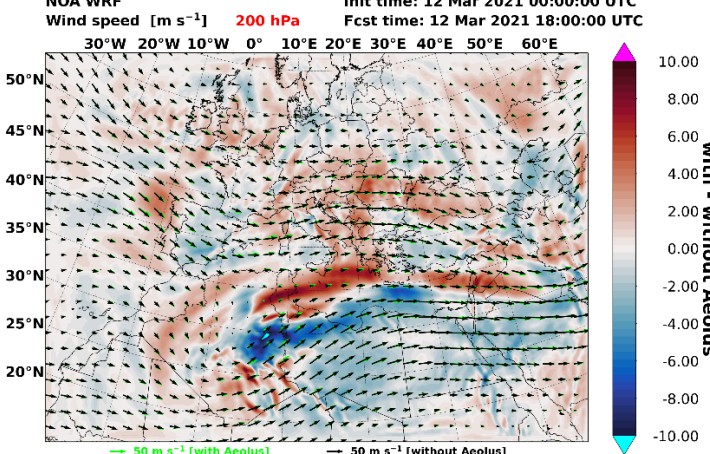

**Figure A 4: Wind speed (m/s) in WRF 18 h forecasts. Horizontal winds (a) "w" Aeolus assimilation, (b) "w/o" Aeolus assimilation and (c) wind speed differences ("w" – "w/o" Aeolus assimilation) at 200 hPa (~ 12 km).**

**(a)**                                                                                                  **(b)**

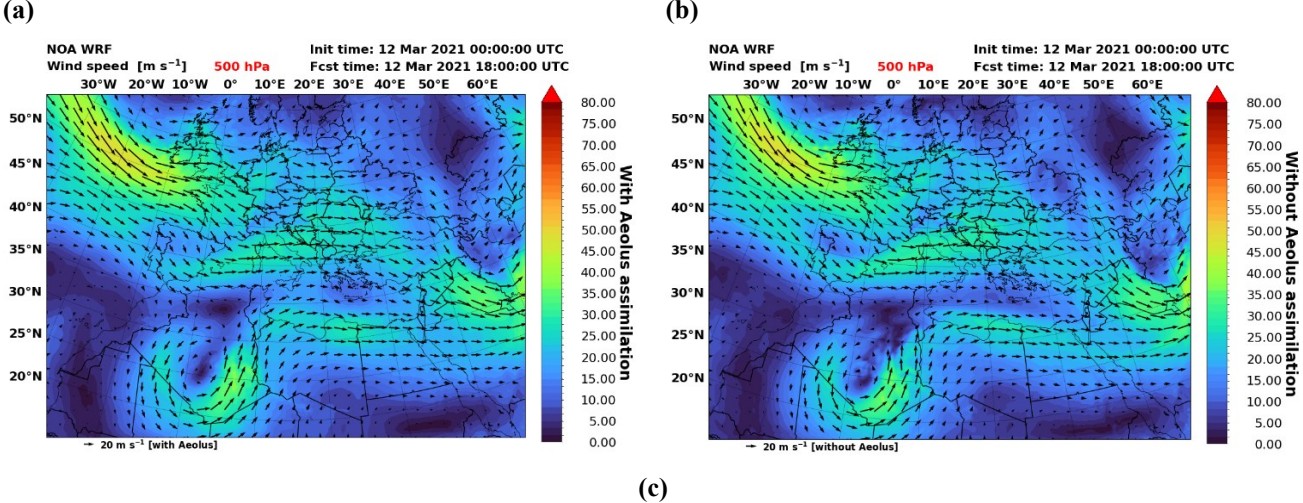

**(c)**

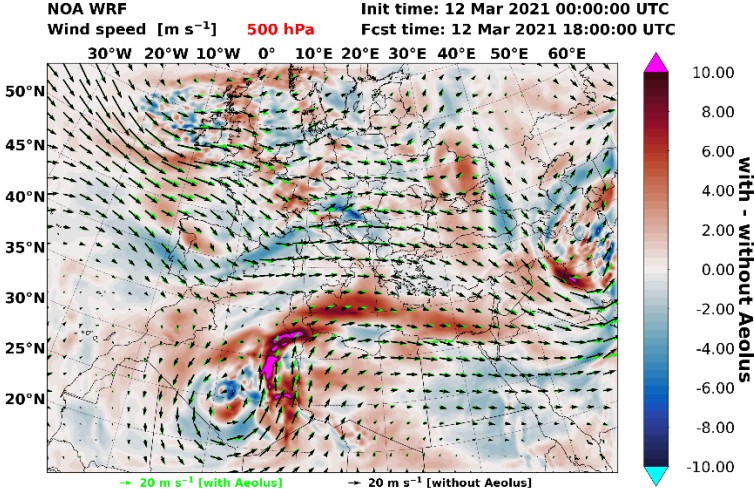

**Figure A 5: Wind speed (m/s) in WRF 18 h forecasts. Horizontal winds (a) "w" Aeolus assimilation, (b) "w/o" Aeolus assimilation and (c) wind speed differences ("w" – "w/o" Aeolus assimilation) at 500 hPa (~ 5.5 km).**

### Code availability

710 The inversion algorithm was written with Python programming language version 3.12 (https:// www. python. org/) and can be obtained from the author Anna Kampouri (akampouri@noa.gr) upon request. The WRF model code is publicly available, has a digital object identifier https:// doi. org/ 10. 5065/ D6MK6 B4K and can be obtained via GitHub (https:// github. com/ wrf-model/ WRF). The FLEXPART-WRF model code is publicly available and can be obtained from https:// www. flexp art. eu/ wiki/ FpOth ermet input. The code used for data processing was written with Python programming language version 3.12
715 (https:// www. python. org/) and can be obtained via GitHub: (https:// github. com/ NOA- ReACT/ Aeolus_Volcano_ 2023). The retrievals and the aerosol lidar optical properties are available from the co-author Anna Gialitaki (togialitaki@noa.gr) upon request.

### Data availability

The Aeolus L2A wind data can be downloaded from: https://apps.ecmwf.int/mars-catalogue/?class=rd&expver=hkv, 720 (ECMWF, 2021) (last access October 2024). The lidar data from the Polly[XT] at PANGEA station (i.e., attenuated backscatter coefficient and volume linear depolarization ratio), were derived using the Single Calculus Chain (SCC; https:// scc. imaa. cnr. it) algorithm; an automatic-analysis tool for lidar data processing, developed within EARLINET (https:// www. earli net. org/) and ACTRIS (https:// www. actris. eu/) and are available by the co-author Anna Gialitaki (togialitaki@noa.gr) upon request. The WRF and FLEXPART-WRF models simulation results are also available by the author Anna Kampouri 725 (akampouri@noa.gr) upon request.

## Author contributions

A.K. conceptualized the manuscript along with V.A., P.Z. and S.So. (Stavros Solomos). All authors wrote parts of the manuscript corresponding to their work and respective results. A.K. performed the FLEXPART and WRF runs along and the inversion algorithm with the support of S.So., P.Z., and T.G.. A.G. and MT, provided the Polly$^{XT}$ lidar retrievals. S.Sc. (Simona Scollo) provided the INGV-OE camera material and synergistic datasets for the analysis. M.R. provided the ECMFW IFS datasets ("w" and "w/o" Aeolus assimilation). All authors provided corrections and suggestions to eventually shape the research, analysis, and the final manuscript. A.K. supervised and directed the whole project.

## Competing interests

The authors declare that they have no conflict of interest.

## Disclaimer

Publisher's note: Copernicus Publications remains neutral with regard to jurisdictional claims in published maps and institutional affiliations.

## Special issue statement

This article is part of the special issue "Aeolus data and their application (AMT/ACP/WCD inter-journal SI)". It is not associated with a conference.

## Acknowledgements

AK and the authors affiliated to the National Observatory of Athens acknowledges the support by the following research projects: the PANGEA4CalVal project funded by the European Union (Grant Agreement 101079201); the e-shape project, under the European Union's Horizon 2020 research and innovation programme (Grant Agreement 820852); The ACTRIS Research Infrastructure; data and services obtained from the PANhellenic GEophysical Observatory of Antikythera (PANGEA) of NOA; Additionally, AK and TG acknowledges support by ESA in the framework of the "Enhancing Aeolus L2A for depolarizing targets and impact on aerosol research and NWP project (4000139424/22/I-NS).

AK acknowledges the support by Dr. Ioanni Binietoglou, Dr. Antoni Gkika and Dr. Emmanouil Proestaki for their invaluable assistance and insightful discussions throughout the development of this work. Additionally, the authors thank the anonymous reviewers for their helpful comments and suggestions improving the quality of the paper.

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
