# Peer review of "Volcanic emission estimates from the inversion of ACTRIS lidar observations and their use for quantitative dispersion modelling"

_EGUsphere, 2024_

## Author Comment (AC1)

We would like to thank the Reviewer for his/her thorough and detailed review. AC replies (regular font) for each comment (bold font) are provided below.

**Review of "Volcanic emission estimates from the inversion of ACTRIS lidar observations and their use for quantitative dispersion modelling"**

**The paper by Kampouri et al. introduces an innovative inversion algorithm incorporating ground-based lidar data to estimate volcanic source emissions for a more accurate ash dispersion modelling. The methodology makes also use of wind data coming from the space wind lidar Aeolus as was first shown by Amiridis et al. (Sci. Rep., 2023). The approach answers to an important issue for the modelling of volcanic hazards for aviation when source information is not complete, thereby bridging research infrastructures and satellite missions. The results highlight the robustness of the technique when compared against ACTRIS ground-based lidar measurements as well as SEVIRI images.**

**The paper reads very well and is appropriate for ACP. However, I would like clarified a few things that will improve the submitted paper. Below I include for consideration specific comments and technical corrections. P1L2 means line 2 of page 1.**

**Specific comments**

- **P1L22 To which "observational data" are you referring?**

**AC.** We would like to thank the reviewer for this comment. In L22, 'observational data' refers to two key datasets used in this study: 1. the ground-based measurements obtained from the Polly$^{XT}$ lidar system at the PANhellenic GEophysical observatory of Antikythera operated by the National Observatory of Athens (PANGEA-NOA from now on) which were used to establish the source-receptor relationships and 2. the volcanic ash plume height observations which were obtained from the ECV calibrated camera operated by the INGV-OE (Calvari et al., 2021; Corradini et al., 2018; Scollo et al., 2019). The latter observations, taken during the time period of 06:30 to 10:30 UTC on March 12th, 2021, were used as a-priori constraints in the ash emissions estimation process. The inversion scheme developed in this study combines a-priori source information with the output of the FLEXPART dispersion model and Polly$^{XT}$ lidar retrievals to estimate volcanic ash emissions more accurately. In the revised text we rephrased the sentence. The sentence now reads "These SRR are then used to derive the emission rates based on observational data including volcanic ash plume heights from INGV-EO and Polly$^{XT}$ lidar retrievals."

- **P1L27 Please mention where the plume was observed.**

**AC.** We would like to thank the reviewer for this comment. In Line 27, the volcanic ash plume was observed at the PANGEA-NOA observatory, using the Polly$^{XT}$ ground-based lidar system. The plume was transported from Etna following the eruption on the 12th of March 2021 and was

detected over the ground station between 18:30 and 21:30 UTC at altitudes ranging from 8 to 12 km above sea level (revised range after more careful calculations. This was also corrected in the revised manuscript). In the revised text we rephrased the sentence which now reads: "Our approach applied to the 12 March 2021 Etna eruption, accurately captures a dense aerosol layer between 8 and 12 km above PANGEA-NOA station."

- **§3 Is it possible to devise a diagram for the methodology? I believe that will be of great help.**

**AC.** In the revised manuscript, we have included Figure A1, which presents a schematic workflow explaining the methodology. We believe this addition will enhance the reader's understanding of our approach. Additionally, we have updated the text in paragraph 3, Line 110, to include the following sentence: "The inverse method employed in this study to estimate volcanic ash emissions integrates a-priori information on ash emissions, ground-based lidar observations, and simulations with a dispersion model, resulting in improved ash emission estimates. Figure A1 presents a schematic workflow outlining the methodology followed in this study, providing a clear overview of the steps involved in our approach. In this section, we describe the datasets and methods employed in the inverse modeling process. "

- **§3 Can the inversion modelling be applied to other volcanoes? Can you upscale?**

**AC.** We would like to thank the reviewer for this insightful question. Yes, the inversion modeling framework developed in this study is not limited to Etna volcano eruptions but can be applied to other volcanoes, provided that suitable observational data are available. The methodology relies on ground-based lidar measurements, dispersion modeling (FLEXPART-WRF), and an inversion algorithm to estimate volcanic ash emissions. Therefore, it can be adapted to different volcanic settings where lidar observations or other remote sensing data like: (i) satellite-based lidars (CALIPSO, EarthCARE), (ii) geostationary satellites (SEVIRI), that are available to constrain the source term. Regarding upscaling, the approach can be extended to a regional or global scale by integrating multiple observation sites from lidar networks such as ACTRIS/EARLINET or incorporating additional satellite data. This would allow for improved ash emission estimates for various volcanic eruptions worldwide. Furthermore, the use of high-resolution wind field data (such as Aeolus or future wind lidar missions) can enhance the accuracy of dispersion forecasts in different geographic regions (possibly lacking sufficient information from radiosondes for example), making the methodology widely applicable for volcanic ash monitoring and forecasting. In the revised text we include a paragraph at the end of Section 3.4 (Inversion Algorithm) mention all the above.

The text now reads: "The inversion scheme presented in this study is not limited to Mt. Etna but can be applied to other volcanic eruptions worldwide, provided that suitable observational data are available. The methodology relies on ground-based lidar measurements, dispersion modeling (FLEXPART-WRF), and an inversion algorithm to estimate volcanic ash emissions. Therefore, it can be adapted to different volcanic settings where lidar observations or other remote sensing data like: (i) satellite-based lidars (CALIPSO, EarthCARE), (ii) geostationary satellites (SEVIRI), that are available to constrain the source term. Additionally, the approach can be extended to a regional or global scale by integrating multiple observation sites from lidar networks such as ACTRIS/EARLINET or incorporating additional satellite data. This would allow for improved ash

emission estimates for various volcanic eruptions worldwide. Furthermore, the use of high-resolution wind field data (such as Aeolus or future wind lidar missions) can enhance the accuracy of dispersion forecasts in different geographic regions (possibly lacking sufficient information from radiosondes), making the methodology widely applicable for volcanic ash monitoring and forecasting."

- **§3.1.1 What if ash/dust coexist? Will this affect the value of the conversion factor? What will be the implications in case of highly polluted areas?**

**AC.** We would like to thank the reviewer for this question. The POLIPHON method which is applied in our study in order to calculate the mass concentrations relies on a combination of lidar and sun-photometer data. Dust and volcanic ash particles' intensive optical properties can be very similar in the sense that both are coarse-mode dominated, non-spherical particles. However volcanic ash particles present higher depolarization ratio values (see for example Figure 1 taken from Groß et al., 2015). In this case, based on the increase in depolarization ratio of the lidar measurements, we could identify the regions where dust and volcanic ash particles co-exist. For the calculation of mass concentration and the extinction-to-mass conversion factor ($c_v$) we utilize information from AERONET observations, namely the vertically integrated particle volume concentration v and the aerosol optical thickness ($\tau$). Currently there is no information in AERONET data to disentangle the contribution of dust and volcanic ash in the coarse mode. Hence in the case where both types of particles co-exist in the atmospheric column, the value of the conversion factor would be affected. Nevertheless, the values used in our study are derived from cases when volcanic ash is unaffected from dust since for the specific case study there is a clear signature of the volcanic ash particles at an altitude of 11-12km, un-affected from dust.

[Figure]

*Figure 1. Layer integrated particle linear depolarization ratio and particle lidar ratio at 355 (left) and 532 nm (right|). Plot is taken from Groß et al., 2015.*

Urban pollution particles represent mainly fine mode aerosols which can be separated both in AERONET measurements and in the lidar data using the depolarization ratio values as done in

POLIPHON method. In that case the conversion factor can be calculated separately for the two aerosol types from $v_{fine}$, $v_{coarse}$ and $\tau_{fine}$, $\tau_{coarse}$ provided from AERONET.

- **P5L133 A 3-hour average period is way too long for decision making purposes. Are you able to perform shorter averages?**

**AC.** We thank the reviewer for making this point and we agree that in case of operational products a 3h averaging period would be too long. Nevertheless, in the present version of our work we are focusing mainly on the demonstration of our methodology, and we do not provide an operational product for aviation safety which could be a future plan. A shorter averaging period is off-course possible depending on a combination of parameters that define the signal to noise ratio of the ground-based lidar system measurements and include 1) the instrument capabilities (i.e. the strength of the laser, the telescope aperture, the background noise in case of daytime data) and 2) the layer optical depth probed by the lidar system.

- **P5L140-142 Please define h**.

**AC.** Done. Where 'h' is the height above ground. In the revised text we changed it.

- **Table 1 I think some discussion is needed for the AERONET input parameters. Is the conversion factor calculated? As the lidar observations were made during night, at what time was the AERONET measurement was made? In case you used literature values, I would like to see the references and clearly stated in the document.**

**AC.** The conversion factors are taken from the literature and were not calculated for the specific case study since the volcanic layer was observed during nigh-time and no photometer retrievals were available. According to the reviewer's suggestion we have now clearly stated the reference for the parameters of Table 1. We also noted a small typo in the $c_v$ value uncertainty for the sulfate particles which is now corrected from 0.04 to 0.02. Added paragraph:

"The lidar ratio of coarse mode volcanic ash at 532 nm is reported to range between 40 and 60 sr in the literature (see for example Groß et al., 2012; table 3 for particle extinction and backscatter values in Floutsi et al., 2023 and Gasteiger et al., 2011). For the fine mode aerosols, we use a mean value of 60 sr following the values reported in the literature for particles of sulphuring nature (see for example Floutsi et al., 2023 and Müller et al., 2007). We also account for a lidar ratio retrieval uncertainty of ~30% to capture the measurement range (Ansmann et al., 2012; Giannakaki et al., 2015; Groß et al., 2013). The particle density values $\rho$ follow from OPAC model for coarse mode mineral component and water soluble component for ash and sulfate particles respectively (Hess et al., 1998; Koepke et al., 2015). For the water soluble component, we assume values at relative humidity of 0% which is considered representative for the altitudes of the volcanic layers. The coarse mode component is not considered as hudrophylic. Finally, the extinction to mass conversion factors $c_v$ are taken from Ansmann et al., (2011) for ash and fine mode particles respectively."

- **P7L208 To what data does the "observed plume height" refer to?**

**AC.** We would like to thank the reviewer for this comment. In L208 'observed plume height' refers to the initial volcanic ash injection height over the Etna summit crater from the VONA reports and field observations, as observed by the INGV observatory. In the revised text, the sentence now reads: "We estimate the a-priori mass eruption rate (MER) for ash particles following Mastin et al., 2009 and Scollo et al., 2019, by inverting the observed plume heights over the Etna summit crater from VONA reports and field observations as observed by the INGV observatory, using the 1-D plume model of Degruyter and Bonadonna, (2012)."

- **P8L211 Why the particle density is different from Table 1?**

**AC.** For the particle density $\rho = 2.6 \pm 0.6$ g/cm$^3$ used in Table 1 we follow the study of Ansmann et al., (2011). The uncertainty in $\rho$ value represents the ambiguity entailed in actual knowledge of the particles composition and their morphology. The mass density of 2.45 g/cm$^3$ in the FLEXPART model gravitational particle settling, assumes a spherical shape based on the work of Näslund and Thaning, (1991). This value differs slightly from the one in Table 1 due to differences in particle shape assumptions, size distributions, and literature sources referenced in various calculations. We have included the following phrase in the manuscript to make this clear to the reader: "The particle density value used in the FLEXPART model (2.45 g/cm$^3$) differs slightly from the density used in Table 1 (2.6 $\pm$ 0.6 g/cm$^3$) due to differences in shape assumptions, size distributions, and literature sources referenced in various calculations."

- **§3.4 Can multiple/concurrent lidar observations be accommodated?**

**AC.** We thank the reviewer for this comment. There is no limitation included in our algorithm regarding the number of lidar observations from different systems that could be accommodated. From a technical point of view this could be even beneficial to constrain the model simulations even more as it would provide more points for the inversion.

- **P9L244-245 Can you confirm that the impossibility of direct retrievals refers to satellite images? If that is not the case, consider volcano radar monitoring. Radar is most sensitive to large particles and can penetrate optically thick plumes near the source.**

**AC.** We would like to thank the reviewer for the question. Yes, passive satellite sensors, while providing near-global coverage and rapid detection of volcanic ash clouds, are inherently limited in their ability to retrieve the vertical distribution of ash within the eruption column. We acknowledge that radar systems are highly sensitive to large volcanic particles and can penetrate dense plumes near the source, offering a valuable complement to optical and lidar-based retrievals. In the revised manuscript we include your comment that could be helpful for future work. The sentence now reads "Ground-based and airborne radar observations, which are sensitive to larger particles and can penetrate optically thick plumes, provide a complementary source of information to retrieve near-source plume properties such as mass eruption rate and column height."

- **P9L253 Please define fine ash as it is the first time it appears in the text.**

**AC.** We would like to thank the reviewer for this suggestion. In the revision text we rephrased the sentence. The sentence now reads "We perform the inversion using a Bayesian approach to provide the best estimate of the emissions profile for fine ash (with particles 3, 5, 9, and 21 μm in diameter) that can …."

- **P11L310 Why does the MER value not coincide with Table 3?**

**AC.** We would like to thank the reviewer for pointing out this inconsistency. Upon reviewing our calculations, we confirm that the MER value of 75,000 kg/s at 10 km was incorrect. The correct value is 58,800 kg/s, as indicated in Table 3. We appreciate the opportunity to correct this error and updated the manuscript accordingly to reflect the accurate MER value.

- **P11L310 How was this height (10 km) estimated?**

**AC.** We would like to thank the reviewer for this question. The height of 10 km (the mean column height from 8:00 to 9:45 UTC, see Table 3) was estimated based on observations from the ECV calibrated camera operated by the INGV-OE. These observations provided consistent estimates of the ash plume height during the peak eruption phase. Additionally, the INGV observatory observed the ash plume from 6:30 to 10:30 UTC (per 15 min). We calculated the average of the injection heights to 6:00 UTC (average from 6:30 to 7:45 UTC), to 8:00 UTC (from 8:00 to 9:45 UTC), and to 10:00 UTC (from 10:00 to 10:30 UTC). The inversion methodology used in our study incorporated these observational constraints to refine the final estimated height. We clarify this in the revised manuscript. Now the text reads (L307-311) "The column heights of the ash plume from the 12 March 2021 were obtained from the ECV calibrated camera operated by the INGV-OE (Calvari et al., 2021; Corradini et al., 2018; Scollo et al., 2019) during the time period of 6:30 to 10:30 UTC. The ash plume height reached up to 9.0 km a.s.l. In order to calculate the a-priori emissions, the data were resampled at ~2-hour intervals, specifically at 6:00 (from 6:30 to 7:45 UTC), 8:00 (from 8:00 to 9:45 UTC) and 10:00 UTC (from 10:00 to 10:30 UTC)."

- **P11L317 It is not clear to me whether you used 11.5 km in your calculations. Please clarify.**

**AC.** We would like to thank the reviewer for this comment. We did incorporate the volcanic Ash Cloud Top Height (ACTH) derived by the SEVIRI satellite for the period between 08:15 and 08:45 UTC, which was estimated at 11.5 km a.s.l. (Calvari et al., 2021). These values have been integrated into our calculations, as presented in the revised version of the manuscript (see Appendix, Table A1 in the revised version). Between 09:00 and 09:30 UTC, the ACTH derived from SEVIRI indicates a gradual decrease in plume height from 10 km to 9.5 km. Additionally, Table A1 presents the ash plume heights as observed from the ECV camera operated by the INGV-OE observatory. However, due to the camera's limited field of view (approximately 9.0–9.5 km a.s.l. Scollo et al., 2014), it was unable to capture the maximum plume elevation during the peak of the eruption. As a result, while the camera data estimated the plume height to be above 9.0 km, we relied on the SEVIRI-derived value of 11.5 km a.s.l. for our calculations during that period.

In the revised text we include the Table A1 in the Appendix and we rephrased the sentence which now reads: "The maximum plume elevation was not recorded by the ECV camera due to its limited

field of view (approx. 9.0 – 9.5 km a.s.l. as noted by Scollo et al., 2014). However, according to SEVIRI aboard the geostationary Meteosat Second Generation satellite, the volcanic Ash Cloud Top Height (ACTH) between 08:15 to 08:45 UTC was estimated at 11.5 km a.s.l. (Calvari et al., 2021). This higher SEVIRI-derived plume height was used into the calculations for the a-priori ash emissions during this time window, as it provides a more accurate representation of the plume height at the peak of the eruption (see Appendix Table A 1)".

*Table A1. Volcanic ash plume heights (m) during the eruption activity from 06:30 to 10:30 UTC, as recorded by the ECV camera operated by INGV-EO (second column) and adjusted heights incorporating SEVIRI satellite observations where applicable (third column).*

| TIME (UTC) | HEIGHT (m) from ECV camera (INGV-EO) | HEIGHT (m) incorporating SEVIRI |
|---|---|---|
| 6:30 | 4000 | 4000 |
| 6:45 | 5500 | 5500 |
| 7:00 | 5500 | 5500 |
| 7:15 | 6000 | 6000 |
| 7:30 | 6500 | 6500 |
| 7:45 | 7000 | 7000 |
| 8:00 | 7500 | 7500 |
| 8:15 | >9000 | 11500 |
| 8:30 | >9000 | 11500 |
| 8:45 | >9000 | 11500 |
| 9:00 | >9000 | 10000 |
| 9:15 | >9000 | 9500 |
| 9:30 | >9000 | 9500 |
| 9:45 | 9000 | 9000 |
| 10:00 | 6500 | 6500 |
| 10:15 | 5000 | 5000 |
| 10:30 | 4500 | 4500 |

- **P12L328 Are you referring to Figure 2b? That figure shows volume linear depolarization ratio, please elaborate.**

**AC.** Thank you for noticing this, now the sentence reads "by the high volume linear depolarization ratios".

- **P12L329 I believe two of the references, apart from Gross et al. (2013), do not quite capture the statement you make. Miffre et al. (2011) use a 355-nm lidar different from the wavelength of the profiles shown here. Also, Pisani et al. (2012) refer to volcanic particles measured at the vent and, therefore, discusses volcanic particles with different characteristics. There are several publications to pick from.**

**AC.** We thank the reviewer. We replaced the Miffre et al. (2011) and Pisani et al. (2012) references in the text with Gasteiger et al., 2011 and Wiegner et al., 2012 which refer to ground-based lidar observations and Tackett et al., 2023 which refers to the CALIPSO stratospheric product climatology (discriminating between volcanic ash, sulfate particles and smoke particles in the stratosphere).

- **P13L346 Why the a-posteriori MER have this behavior? Any ideas?**

**AC.** We would like to thank the reviewer for this insightful question. The inversion process incorporates ground-based lidar observations from Polly$^{XT}$ lidar system, which provide direct measurements of the volcanic ash plumes vertical structure. These lidar observations, being highly sensitive to the vertical distribution of ash, help refine the emission estimates, particularly within the altitude range where the plume was observed (8–12 km), ensuring that the final estimates better reflect the actual emission patterns.

Additionally, the a-priori MER estimates were initially derived from empirical relationships based on VONA-reported plume heights (see Appendix Table A1). These empirical methods often overestimate emissions, particularly in the early stages of the eruption when plume dynamics are highly variable. The inversion process corrects for this by adjusting emissions to match the lidar observations, leading to lower and more consistent a-posteriori MER values. As a result, the a-posteriori MER shows a more physically realistic and concentrated vertical distribution, predominantly between 8 and 12 km, in contrast to the a-priori estimates, which exhibit a wider spread, including lower altitudes where the plume was not actually observed. This suggests that the inversion effectively filters out unrealistic emissions, yielding a more refined and accurate vertical profile.

Moreover, it is important to highlight that during the eruption, the plume undergoes a dynamic evolution, rising and collapsing at different stages. This complex evolution is not fully captured by the model, which does not explicitly simulate the transient rise and fall of the plume. Instead, the model adjusts the a-posteriori MER at each height over time, increasing or decreasing emission rates to best match the available observations. While this approach provides a more constrained and consistent representation of emissions, some rapid fluctuations in plume height and intensity may not be fully resolved within the model framework. Finally, we emphasize that the most significant refinement in the inversion algorithm occurs between 08:00–08:45 UTC, at the peak of the eruption.

In the revised manuscript we added a paragraph to clarify better the MER behavior. The text now reads "The eruption dynamics involve a complex evolution of the volcanic plume, with phases of rising and collapsing. However, this dynamic behavior is not explicitly resolved in the a-posteriori simulations, which do not capture rapid fluctuations in plume height and intensity. Instead, the inversion algorithm adjusts the a-posteriori MER at each altitude over time, dynamically increasing or decreasing emission rates to achieve the best agreement with available observations. The most significant refinement occurs between 08:15–08:45 UTC, within the 8–12 km altitude range, where lidar observations provide direct constraints on the plume's vertical structure. As a result, the

inversion optimizes the emission estimates primarily within this altitude range, ensuring the highest degree of agreement between observed and a-posteriori emissions."

- **Figure 3 Why not make the time axis consistent between b) and c)?**

**AC.** We would like to thank the reviewer for this suggestion. The time axis in Figure 3b and 3c is consistent. However, the colorbar in Figure 3b reduces the plot size, which may give the impression of inconsistency. In the revised manuscript, we have adjusted the plot size in Figure 3b to improve visual consistency.

- **P15L402-403 Considering that any orbiting lidar offers a few opportunities for your approach, is this a weakness of the methodology? What is your feeling?**

**AC.** We would like to thank the reviewer for this insightful question. Indeed, the limited overpass frequency of polar orbiting satellites equipped with lidar systems, presents a constraint for our methodology, as it reduces the number of opportunities to directly observe volcanic ash plumes. However, we do not see this as a weakness but rather as a limitation that should be considered when applying the approach. The strength of our methodology lies in its ability to integrate available lidar aerosol and wind observations with dispersion modeling, allowing us to derive improved emission estimates even from a limited number of overpasses. However, the methodology could be enhanced by the utilization of ground-based lidars.

- **Figure 5 I think the lidar map would help the interpretation.**

**AC.** We thank the reviewer for this suggestion. If by "lidar map" the reviewer is referring to Figure 2, we believe that maintaining the current structure of the results provides a clearer distinction between observations and model outputs. Integrating Figure 2 into Figure 5 may not significantly enhance the interpretation and could disrupt the logical flow of the analysis.

- **P16L414-417 This result seems important to be worth mentioning in the paper's abstract.**

**AC.** We would like to thank the reviewer for this suggestion. In the revision text we rephrased the abstract. The abstract now reads "Results show a minimal difference of the order of 2 % between the observed and the simulated ash concentrations. Furthermore, the structure of the a-posteriori ash plume closely resembles the ash cloud image captured by the SEVIRI satellite above Antikythera island, highlighting the novelty of the inversion results."

- **P17L428 The reported maximum height seems higher to my eye than that seen on Figure 5. Is it correct?**

**AC.** We would like to thank the reviewer for this comment. We confirm that the reported maximum height is 12 km, not 12.5 km as previously stated in the text. The FLEXPART model represents heights as the average altitude of each layer. To improve clarity, we have revised the plots in Figure 5 with a higher resolution and have adjusted the plotted concentration range to 5–13 km, aligning with the height range observed in the Polly$^{XT}$ lidar quicklook (Figure 2). The a-posteriori layer now

clearly extends from 8 to 12 km. The revised manuscript includes the updated figure to enhance accuracy and visualization.

- **P18L453 Is the a-posteriori "w/o" Aeolus simulation zero? Or wasn't it possible to estimate? Please, rephrase analogously. Also, if the a-posteriori "w/o" Aeolus simulation is zero, what are the implications for the inversion scheme?**

**AC.** We would like to thank the reviewer for this question. The volcanic plume in the "w/o" Aeolus simulation never reached Antikythera on 12 March 2021 between 18:00-21:30 UTC, the time during which the Polly$^{XT}$ lidar system observed volcanic particles above the station. This is due to the fact that the forecasted cloud was shifted to the north, as illustrated in Figure 4a. Consequently, the source-receptor sensitivities derived from the FLEXPART model using the "w/o" Aeolus assimilated wind fields indicate that the volcanic particles arriving above the PANGEA station at altitudes of 8–10 km were few and originated from release heights around 8–11 km above Etna, but only in the simulation where the particle release time was 04:00-06:00 UTC (Figure A2, right panel). However, this release time does not align with actual eruption observations, as the event commenced at 06:00 UTC according to the VONA messages from INGV-EO. Additionally, the SRR had zero values in the simulations where the release time of the particles was 06:00-08:00 and 08:00-10:00 UTC.

Since the source-receptor matrix (SRM) is a fundamental component of the inversion algorithm, and in the "w/o" Aeolus simulation this matrix was effectively zero, it was not possible to calculate the a-posteriori MER in the "w/o" Aeolus stimulation. This result underscores the importance of accurate wind field assimilation for robust source-term estimations in volcanic ash inversion modeling. To clarify this in the revised manuscript, we have modified the text as follows:

"The a-priori and a-posteriori ash mass concentrations "w/o" Aeolus simulation equals to zero and are not shown, as the SRM derived using 'w/o' Aeolus assimilated wind fields results in negligible sensitivities. The volcanic ash plume in the 'w/o' Aeolus simulation never reached Antikythera on 12 March 2021 between 18:00-21:30 UTC due to a northward shift (Figure 4a)."

- **Figure 7 I suppose that Figures 6 and 7 can be presented together. I think that "a-" is needed for priori/posteriori in the legend. Also, in the legend, state that the "A-priori "w" Aeolus" is a result of Amiridis et al. (2023). In the same style, "this study" can preplace "A-posteriori "w" Aeolus".**

**AC.** We would like to thank the reviewer for the suggestion to present Figures 6 and 7 together. We have implemented this change in the revised manuscript (where Figure 7 is now Figure 6d). Additionally, we have updated the figure legend to include 'a-' for the priori/posteriori distinction. Regarding the terminology, we prefer to keep "a-posteriori," as inversion results in previous studies (e.g., Eckhardt et al., 2008; Kristiansen et al., 2014; Stohl et al., 2011) also refer to them in this manner, ensuring consistency with established literature. Furthermore, we have clarified in the Caption that the "a-priori 'w' Aeolus" results are based on Amiridis et al. (2023).

- **§5 Prompt data delivery is crucial for real-time warnings for aviation. For the considered inversion scheme, how long will it take to deliver data? Can it be part of an automatic procedure?**

**AC.** We thank the reviewer for the question. It can be part of automated procedure, since all the involved datasets (meteorological datasets and remote sensing observations) are available in near real time. Furthermore, the inversion algorithm is computationally efficient and can process each simulation in few minutes.

- **P20L491-493 Isn't this a finding of Amiridis et al. (2023)?**

**AC.** We thank the reviewer for this comment. Yes, the result according to the vertical shift of 1 km between the observed and simulated ash mass concentration peaks is a result in Amiridis et al., 2023. In the revised text we add the reference.

- **P20L497-499 As I already mentioned, it would be nice to highlight the good performance of the inversion scheme with independent measurements as shown in figure 4. If I can take it a bit further, does the comparison of the model with the lidar profiles provide a "fair" comparison? I say this because the lidar profiles were already used in your methodology. I believe that a second lidar somewhere along the path of the plume would provide an ideal dataset to compare to.**

**AC.** We thank the reviewer for this valuable suggestion. We agree that an independent dataset, such as ground-based, satellite remote sensing data, or airborne in-situ measurements between the source and the receptor would provide an even more reliable validation for this methodology. Unfortunately, in that case we do not have other available vertically resolved ground-based (i.e. lidar data) or airborne in-situ datasets, apart from the SEVIRI satellite image presented in Figure 4. We will incorporate this important point in the revised manuscript, highlighting the need for such validation in future work. The text now reads "This consistency highlights the robustness of the new inversion algorithm and the significant improvement in the vertical distribution and the ash mass concentration. However, additional independent datasets, such as ground-based, satellite remote sensing data, or airborne in-situ measurements along the plume's trajectory, would further enhance the validation of this methodology and should be considered in future studies."

**Technical Corrections**

- **P1L18 Add "," before "e.g.,".**

**AC.** Done

- **P1L21 Remove "ensuing".**

**AC.** Done

- **P1L23 Please give the acronym or capital letters are unnecessary.**

**AC.** We would like to thank the reviewer for this comment. Done

- **P1L25 Move "evolution" after the "of the plume's" and add "the" before "wind fields".**

**AC.** Done

- **P2L45 Please add "usually" before "they do not". VAAC reports during major eruptions might contain quantitative information.**

**AC.** Done

- **P3L73 Please follow the Journal's terminology on dates.**

**AC.** We would like to thank the reviewer for this comment. We have also changed the date format in the manuscript, according to the journal's guidelines, in Lines: 84, 94, 104 (Fig. 1), 185, 204, 302 321, 332 (Fig. 2), 340, 371-72 (Fig. 3), 388, 390, 392 (Fig.4), 405 (Fig. 5), 415, 444 (Fig. 6), 471 (Fig. 7), 479, 526, 539 (Fig. A1).

- **P3L74 Remove parentheses. Replace ";" with "and". Please take extra care when reporting literature references as I noticed several inconsistencies throughout the manuscript. In the following, I report the ones that I found.**

**AC.** We would like to thank the reviewer for the valuable comments and careful review of the manuscript. We carefully addressed reviewers' suggestions by thoroughly reviewing and correcting inconsistencies in literature references throughout the manuscript.

- **P3L84 Replace "sulfate" with "SO2".**

**AC.** Done

- **P4L114 Remove "location".**

**AC.** Done

- **P4L116 Replace "megacities" with "cities".**

**AC.** Done

- **P5L118: Replace "lidar system the type of PollyXT" with "PollyXT lidar system".**

**AC.** Done

- **P5L127 Replace "ang" with "and".**

**AC.** Done

- **P5L129 Remove the hyperlink.**

**AC.** Done

- **P6L164 Remove "placed".**

**AC.** Done

- **P6L170 "maintaining… medium range" What do you mean?**

**AC.** We would like to thank the reviewer for this comment. We mean that the improvement in wind forecasts (0.5–2% in terms of root-mean-square error) continues to have a noticeable effect not only in the short term but also in the medium-range forecast period (which typically

refs to weather predictions from about 3 to 10 days ahead). In the revision text we rephrased the sentence. The sentence now reads "The improvement in wind forecasts ranges from 0.5 % to 2 %, in terms of root-mean-square error, maintaining a significant impact even into the medium range weather forecasting."

- **P7L179 Remove "had".**

**AC.** Done

- **P7L195 Remove "being".**

**AC.** Done

- **P7L202 Replace "allow" with "allows".**

**AC.** Done

- **P7L206 Replace "… a vertical resolution 1 km in the range extending…" with "…1 km vertical resolution…"**

**AC.** Done

- **P7L207 Remove parentheses and correct references accordingly.**

**AC.** Done

- **P7L208-209 Remove parentheses and add an opening parenthesis to "2012)".**

**AC.** Done

- **P8L225 Replace ", each one being" with "of" and replace "thick" with "thickness".**

**AC.** Done. We would like to thank the reviewer for this suggestion.

- **Table 2 Remove "M.J."**

**AC.** Done. We have corrected the reference "Iacono, M.J. et al., 2008" to "Iacono, et al., 2008".

- **P9L237 Remove parentheses and correct references accordingly.**

**AC.** Done

- **P9L242 Remove "mass eruption rate" and the parentheses from "(MER)".**

**AC.** Done

- **P9L247 Replace "near the source" with "downwind".**

**AC.** Done.

- **P9L248 Can you rephrase the second point of the list?**

**AC.** Done. We rephrased the second point of the point list. The sentence now reads "ii) the integration of Aeolus meteorological wind fields (ECMWF, 2021) into the FLEXPART-WRF model".

- **P9L254-255 Please rephrase "that can undergo long-range dispersion".**

**AC.** Done. In the revision text we rephrased the sentence. The sentence now reads "We perform the inversion using a Bayesian approach to provide the best estimate of the emissions profile for fine ash that can be transported over long distances."

- **P10L273 Remove full stop before the opening parenthesis and replace "are further described in" with "see").**

**AC.** Done.

- **P10L290 Enclose Eq. (9) in parentheses.**

**AC.** Done. We have done the same for Eq. (8) in Line 287.

- **P11L298 Remove "mass eruption rate" and the parentheses.**

**AC.** Done.

- **P11L299-300 Remove the parentheses from the first reference and enclose the second reference in parentheses.**

**AC.** Done

- **P11L307 Replace "formatted" with "formed".**

**AC.** Done. The sentence now reads "a stronger plume formed".

- **P11L316 Remove "Simona".**

**AC.** Done. We have removed Simona from the reference. The reference now is "Scollo et al., 2014".

- **P12L322 The cited papers already appear in Section 3.1.**

**AC.** Done. We have removed the references from L322, as they already appear in Section 3.1.

- **P12L325 Remove "during this period".**

**AC.** Done.

- **P12L326 Remove "being".**

**AC.** Done.

- **P12L338 Remove the second ",".**

**AC.** Done. We have removed the second "," The sentence now reads "outlined by Scollo et al. (2019)".

- **P13L342-343 Remove "on 12 March 2021, between 06:30 and 10:30 UTC".**

**AC.** Done. The sentence now reads "The a-priori MER was obtained by inverting observed plume heights from the VONA reports, based on data collected by calibrated cameras operated by the INGV-EO observatory."

- **P13L362 Remove "Simona".**

**AC.** Done

- **P13L364 Remove the second "," and "in their results".**

**AC.** Done. The sentence now reads "Additionally, Calvari et al. (2021) further indicate that the observed plume column altitudes predominantly range between 6 and 9 km, which is the upper limit of the INGV-OE camera system."

- **P17L426 Remove " "w" ".**

**AC.** Done.

- **P17L435 Replace "This enhancement" with "The better agreement".**

**AC.** Done. The sentence now reads "The better agreement in both the vertical distribution..."

- **P20L480 Remove "experiment".**

**AC.** Done. We have removed "experiment". And we added "simulation". The sentence now reads "the simulation was repeated twice".

- **P20L481-482 What do you mean by "along with… following the eruption"?**

**AC.** We thank the reviewer for this comment. We mean that the cloud-free conditions observed above PANGEA station, the days after the eruption, provided an ideal context for studying the volcanic aerosol layers. These conditions allowed for clearer observations and better analysis of aerosol behavior. We rephrase the sentence. The sentence now reads "The volcanic aerosol layers observed above the PANGEA-NOA station in Antikythera, along with the clear sky conditions in the days after the eruption, made this an ideal test case."

**References**

Ansmann, A., Tesche, M., Seifert, P., Groß, S., Freudenthaler, V., Apituley, A., Wilson, K. M., Serikov, I., Linné, H., Heinold, B., Hiebsch, A., Schnell, F., Schmidt, J., Mattis, I., Wandinger, U., and Wiegner, M.: Ash and fine-mode particle mass profiles from EARLINET-AERONET observations over central Europe after the eruptions of the Eyjafjallajökull volcano in 2010, J. Geophys. Res. Atmospheres, 116, https://doi.org/10.1029/2010JD015567, 2011.

Ansmann, A., Seifert, P., Tesche, M., and Wandinger, U.: Profiling of fine and coarse particle mass: case studies of Saharan dust and Eyjafjallajökull/Grimsvötn volcanic plumes, Atmospheric Chem. Phys., 12, 9399–9415, https://doi.org/10.5194/acp-12-9399-2012, 2012.

Calvari, S., Bonaccorso, A., and Ganci, G.: Anatomy of a Paroxysmal Lava Fountain at Etna Volcano: The Case of the 12 March 2021, Episode, Remote Sens., 13, 3052, https://doi.org/10.3390/rs13153052, 2021.

Degruyter, W. and Bonadonna, C.: Improving on mass flow rate estimates of volcanic eruptions, Geophys. Res. Lett., 39, 2012GL052566, https://doi.org/10.1029/2012GL052566, 2012.

Eckhardt, S., Prata, A. J., Seibert, P., Stebel, K., and Stohl, A.: Estimation of the vertical profile of sulfur dioxide injection into the atmosphere by a volcanic eruption using satellite column measurements and inverse transport modeling, 2008.

ECMWF: ECMWF Starts Assimilating Aeolus Wind Data., 2021.

Floutsi, A. A., Baars, H., Engelmann, R., Althausen, D., Ansmann, A., Bohlmann, S., Heese, B., Hofer, J., Kanitz, T., Haarig, M., Ohneiser, K., Radenz, M., Seifert, P., Skupin, A., Yin, Z., Abdullaev, S. F., Komppula, M., Filioglou, M., Giannakaki, E., Stachlewska, I. S., Janicka, L., Bortoli, D., Marinou, E., Amiridis, V., Gialitaki, A., Mamouri, R.-E., Barja, B., and Wandinger, U.: DeLiAn – a growing collection of depolarization ratio, lidar ratio and Ångström exponent for different aerosol types and mixtures from ground-based lidar observations, Atmospheric Meas. Tech., 16, 2353–2379, https://doi.org/10.5194/amt-16-2353-2023, 2023.

Gasteiger, J., Groß, S., Freudenthaler, V., and Wiegner, M.: Volcanic ash from Iceland over Munich: mass concentration retrieved from ground-based remote sensing measurements, Atmospheric Chem. Phys., 11, 2209–2223, https://doi.org/10.5194/acp-11-2209-2011, 2011.

Giannakaki, E., Pfüller, A., Korhonen, K., Mielonen, T., Laakso, L., Vakkari, V., Baars, H., Engelmann, R., Beukes, J. P., Van Zyl, P. G., Josipovic, M., Tiitta, P., Chiloane, K., Piketh, S., Lihavainen, H., Lehtinen, K. E. J., and Komppula, M.: One year of Raman lidar observations of free-tropospheric aerosol layers over South Africa, Atmospheric Chem. Phys., 15, 5429–5442, https://doi.org/10.5194/acp-15-5429-2015, 2015.

Groß, S., Freudenthaler, V., Wiegner, M., Gasteiger, J., Geiß, A., and Schnell, F.: Dual-wavelength linear depolarization ratio of volcanic aerosols: Lidar measurements of the Eyjafjallajökull plume over Maisach, Germany, Volcan. Ash Eur. Erupt. Eyjafjallajöekull Icel. April-May 2010, 48, 85–96, https://doi.org/10.1016/j.atmosenv.2011.06.017, 2012.

Groß, S., Esselborn, M., Weinzierl, B., Wirth, M., Fix, A., and Petzold, A.: Aerosol classification by airborne high spectral resolution lidar observations, Atmos Chem Phys, 13, 2487–2505, https://doi.org/10.5194/acp-13-2487-2013, 2013.

Groß, S., Freudenthaler, V., Schepanski, K., Toledano, C., Schäfler, A., Ansmann, A., and Weinzierl, B.: Optical properties of long-range transported Saharan dust over Barbados as measured by dual-wavelength depolarization Raman lidar measurements, Atmospheric Chem. Phys., 15, 11067–11080, https://doi.org/10.5194/acp-15-11067-2015, 2015.

Hess, M., Koepke, P., and Schult, I.: Optical Properties of Aerosols and Clouds: The Software Package OPAC, Bull. Am. Meteorol. Soc., 79, 831–844, https://doi.org/10.1175/1520-0477(1998)079<0831:OPOAAC>2.0.CO;2, 1998.

Koepke, P., Gasteiger, J., and Hess, M.: Technical Note: Optical properties of desert aerosol with non-spherical mineral particles: data incorporated to OPAC, Atmospheric Chem. Phys., 15, 5947–5956, https://doi.org/10.5194/acp-15-5947-2015, 2015.

Kristiansen, N. I., Arnold, D., and Stohl, A.: Procedures for volcano inversions using FLEXPART, 2014.

Mastin, L. G., Guffanti, M., Servranckx, R., Webley, P., Barsotti, S., Dean, K., Durant, A., Ewert, J. W., Neri, A., Rose, W. I., Schneider, D., Siebert, L., Stunder, B., Swanson, G., Tupper, A., Volentik, A., and Waythomas, C. F.: A multidisciplinary effort to assign realistic source parameters to models of volcanic ash-cloud transport and dispersion during eruptions, J. Volcanol. Geotherm. Res., 186, 10–21, https://doi.org/10.1016/j.jvolgeores.2009.01.008, 2009.

Müller, D., Ansmann, A., Mattis, I., Tesche, M., Wandinger, U., Althausen, D., and Pisani, G.: Aerosol-type-dependent lidar ratios observed with Raman lidar, J. Geophys. Res. Atmospheres, 112, https://doi.org/10.1029/2006JD008292, 2007.

Näslund, E. and Thaning, L.: On the Settling Velocity in a Nonstationary Atmosphere, Aerosol Sci. Technol., 14, 247–256, https://doi.org/10.1080/02786829108959487, 1991.

Scollo, Michele Prestifilippo, Emilio Pecora, Stefano Corradini, Luca Merucci, Gaetano Spata, and Mauro Coltelli: Eruption column height estimation of the 2011-2013 Etna lava fountains, Ann. Geophys., 57, 3, https://doi.org/10.4401/ag-6396, 2014.

Scollo, S., Prestifilippo, M., Bonadonna, C., Cioni, R., Corradini, S., Degruyter, W., Rossi, E., Silvestri, M., Biale, E., Carparelli, G., Cassisi, C., Merucci, L., Musacchio, M., and Pecora, E.: Near-Real-Time Tephra Fallout Assessment at Mt. Etna, Italy, Remote Sens., 11, 2987, https://doi.org/10.3390/rs11242987, 2019.

Stohl, A., Prata, A. J., Eckhardt, S., Clarisse, L., Durant, A., Henne, S., Kristiansen, N. I., Minikin, A., Schumann, U., Seibert, P., Stebel, K., Thomas, H. E., Thorsteinsson, T., Tørseth, K., and Weinzierl, B.: Determination of time- and height-resolved volcanic ash emissions and their use for quantitative ash dispersion modeling: the 2010 Eyjafjallajökull eruption, Atmospheric Chem. Phys., 11, 4333–4351, https://doi.org/10.5194/acp-11-4333-2011, 2011.

Tackett, J. L., Kar, J., Vaughan, M. A., Getzewich, B. J., Kim, M.-H., Vernier, J.-P., Omar, A. H., Magill, B. E., Pitts, M. C., and Winker, D. M.: The CALIPSO version 4.5 stratospheric aerosol subtyping algorithm, Atmospheric Meas. Tech., 16, 745–768, https://doi.org/10.5194/amt-16-745-2023, 2023.

Wiegner, M., Gasteiger, J., Groß, S., Schnell, F., Freudenthaler, V., and Forkel, R.: Characterization of the Eyjafjallajökull ash-plume: Potential of lidar remote sensing, Phys. Chem. Earth Parts ABC, 45–46, 79–86, https://doi.org/10.1016/j.pce.2011.01.006, 2012.

---

## Author Comment (AC2)

**AC2 to RC2 Anonymous Referee #1**

We would like to thank the Reviewer for his/her thorough and detailed review. AC replies (regular font) for each comment (bold font) are provided below.

**General Comments: (overall quality)**

The manuscript integrates the ground-based and satellite observational data with atmospheric transport models (FLEXPART) to demonstrate the impact of volcanic eruptions on different aerosol concentrations observed at different ground-based atmospheric monitoring stations. The inversion model developed in this manuscript provides an innovative solution for developing a volcanic hazard warning model for aviation when source information is incomplete, synergistically bridging ACTRIS research infrastructures, satellite missions and modeling. Although only a study is presented that targets the eruption of the Etna volcano on March 12, 2021, the results obtained demonstrate the capability of the model to estimate volcanic ash emissions from lidar and FLEXPART data.

The manuscript is quite clean from an editorial point of view, well structured and is appropriate for ACP. However, a few questions should be addressed, and I would appreciate further discussion in the manuscript.

**Specific comments:**

**1. Page 2, Lines 60-65 (Introduction): I recommend that you add a phrase or two to improve the description of the volcanic particles radiative effects, both direct and indirect. Also, you must include some key references for that.**

AC. We would like to thank the reviewer for this suggestion. In the revised manuscript we add this paragraph "Moreover, volcanic particles can influence the planetary radiative balance through both direct and indirect effects, introducing significant uncertainties in plume dispersion and lifetime. The direct effect involves the scattering and absorption of solar and terrestrial radiation, where fine ash and sulfate aerosols contribute to surface cooling or atmospheric warming depending on particle composition, size distribution, and injection plume height (Robock, 2000; Sicard et al., 2025). The indirect effect relates to the role of volcanic particles in cloud micro- and macrophysical properties. Ash particles can act as cloud condensation nuclei (CCN), facilitating water droplet formation and, under specific pressure and temperature conditions, as ice nuclei (IN) (Guerrieri et al., 2023). These processes can alter cloud optical and microphysical properties, enhance cloud reflectivity, affect cloud lifetimes and increase the uncertainties in radiative transfer. Additionally, volcanic ice clouds can hide possible ash layers and pose a severe threat to aviation safety. Atmospheric transport models often struggle to account for these complex interactions, leading to uncertainties in plume evolution, trajectory forecasts, and deposition estimates. Furthermore, the absence of significant physical processes and dependence on empirical relations and data from previous eruptions further contributes to substantial uncertainties in estimates of the erupted mass."

2. Page 3, Lines 75-79 (Introduction): Do you know of previous studies on volcanic particles (volcanic ash, sulfate) that have advanced the integration of data from remote sensing measurements and atmospheric transport modeling? I recommend that you provide an

**overview of what has been done before in terms of integrating observational data from ground, satellite and atmospheric transport models.**

**AC.** We appreciate the reviewer's suggestion. In the revised manuscript, we have included an overview of previous studies that have advanced the integration of remote sensing observations (both ground-based and satellite instruments) with atmospheric transport models for volcanic ash and sulfate aerosol dispersion. The revised text is incorporated into the Introduction as follows:

[revised manuscript text omitted]

3. Page 4, Lines 95-102 (The Case of 12 March–14 March 2021 Etna Volcanic Eruption): The authors should show the relevant meteorological maps for the period with significant volcanic activity from February - March 2021 (500, 300, 200 and 100 hPa circulation) to support the conclusions on volcanic particles transport. Even if FLEXPART ingests the upper air data from ECMWF, a cross-validation with "real meteorological data" (including AEOLUS data) will make the case more convincing. Consider adding a paragraph to the results section to further explore the transport and dispersion of volcanic ash particles.

**AC.** We thank the reviewer's valuable suggestion. We add the following paragraph to the results section. Unfortunately, we were unable to extend model simulations to earlier periods due to data availability constraints, as the European Centre for Medium-Range Weather Forecasts (ECMWF) provided data only for this specific timeframe (from 12 to 14 March 202). These datasets were produced explicitly to investigate the potential improvements attributed to the assimilation of Aeolus wind profiles.

"The transport and dispersion of volcanic ash particles are strongly influenced by upper-air circulation patterns, which play a crucial role in determining the trajectory and lifetime of the volcanic plume. To assess the sensitivity of the volcanic ash transport to the driving meteorology, two simulations were performed using the WRF regional model over the study period. These simulations were driven by two versions of the ECMWF-IFS global model: one incorporating Aeolus wind profile assimilation ("w") and one without Aeolus assimilation ("w/o") (see Sect. 3.3). To evaluate the influence of upper-level circulation on volcanic ash transport, wind maps at 100, 200, 300, and 500 hPa, were generated, for the period of significant volcanic activity (Figure A 3-A 5).

Given that lidar observations estimated the volcanic plume's center of mass at approximately 10 km, the analysis primarily focused on the 300 hPa level (~9.6 km), which closely corresponds to this altitude (Figure 4). Analyzing the WRF regional model wind vectors at upper-tropospheric levels (300 hPa, ~9.6 km) at 18:00 UTC (approximately 11 hours after the Etna eruption), the general atmospheric circulation remained predominantly zonal over the Mediterranean, with westerly winds prevailing throughout the troposphere. Over the Anatolian Plateau and the Eastern Mediterranean Sea, these winds transition into northwesterlies, favoring the direct transport of the Etna plume towards Greece and the Eastern Mediterranean.

A comparison of the two simulations ("w" and "w/o" Aeolus assimilation) indicates that the overall atmospheric pattern remains consistent, with the subtropical and polar jet streams dominating the circulation. However, notable differences in wind speed are evident, as highlighted in the wind speed difference map for the WRF 18-hour forecast (Figure 4).

The color shading in Figure 4 (c) illustrates the differences between the two WRF runs on 12 March 2021 (18:00 UTC). This comparison indicates significant strengthening of winds at 300 hPa when

Acolus wind profiles are assimilated (Figure 4 c), with maximum difference values reaching approximately 8 m/s. Additionally, slight differences in wind vector direction ("w" Acolus (green) and "w/o" Acolus (black)) are observed, particularly over the Ionian Sea (from W to NW) and the Eastern Mediterranean between Crete and Cyprus (from WNW to NW), where the two jet streams merge.

Similar wind speed tendencies are observed at 200 hPa (Figure A 4). In contrast, at 500 hPa (Figure A 5), the influence of Aeolus assimilation is less pronounced, indicating that the most significant differences occur at higher altitudes where jet stream dynamics dominate.

At 100hPa (Figure A 3), a strong westerly jet stream is evident across Europe and North Africa, indicating fast-moving winds that could contribute to long-range transport of volcanic particles. The corresponding wind speed difference map (Figure A 3 c) shows high differences mainly along the jet stream axis, suggesting that Aeolus assimilation plays a crucial role in improving the representation of high-altitude wind fields critical for long-range ash transport.

These findings highlight the importance of accurate upper-air circulation representation in volcanic ash transport modeling. The inclusion of Aeolus wind profiles in the ECMWF-IFS model leads to a more refined depiction of wind patterns, particularly at upper tropospheric and lower stratospheric levels, which are crucial for accurately forecasting the dispersion of volcanic emissions."

Regarding the second part of the comment a cross-validation with "real meteorological data", in this study, the meteorological simulations are conducted using the WRF-ARW model, which relies on initial and boundary conditions derived from two versions of ECMWF-IFS data. One version incorporates assimilated Aeolus Rayleigh-clear and Mie cloudy horizontal line-of-sight (HLOS) L2B wind profiles (the "w" Aeolus experiment), while the other excludes Aeolus data (the "w/o" Aeolus experiment). These meteorological fields subsequently drive the FLEXPART simulations, ensuring a comprehensive representation of atmospheric transport mechanisms.

The generated wind maps from both simulations (with and without Aeolus assimilation) indicate that the wind direction in both cases facilitates the long-range transport of volcanic particles from Etna to Greece (and more specific to Antikythera). This result underscores the importance of upper-level circulation in volcanic ash dispersion and supports the study's conclusions regarding its role in atmospheric transport processes.

The IFS analysis with and without Aeolus assimilation is considered good enough for the objectives of this study. The primary focus is the development of the inversion algorithm using ground-based data, rather than the verification of the IFS wind fields. Since Aeolus wind data are already incorporated into the meteorological fields, the study inherently accounts for their impact on atmospheric circulation. Consequently, a detailed verification of the IFS data is beyond the scope of this work.

**4. Page 4, Lines 113-117 (Methods and Data): The authors should include a description of the general aspects of climate and atmospheric synoptic scale circulation for the region where is located PANGEA-NOA observatory.**

**AC.** We thank the reviewer for the comment. We have now included a description of the general aspects of climate and synoptic-scale atmospheric circulation in the Eastern Mediterranean region.

"The Mediterranean region, particularly its Eastern basin, serves as a confluence of air masses originating from Europe, Asia, and Africa. In this region, anthropogenic emissions from large urban centers interact with natural emissions from the Saharan and Middle Eastern deserts, smoke from frequent wildfires, and volcanic particles from eruptions, notably from Mt. Etna and Icelandic volcanoes. Additionally, the atmosphere over the Eastern Mediterranean contains background marine aerosols and pollen particles from oceanic and vegetative sources. Aerosols exert a variety of effects on regional weather and climate, impacting solar radiation, visibility, and human health, and they pose significant concerns for aviation safety (WMO, 2024).

The Eastern Mediterranean is characterized by a Mediterranean climate, with hot, dry summers and mild, wet winters. This seasonal variability is driven primarily by the interaction between midlatitude westerlies and subtropical high-pressure systems (Lensky et al., 2018). During winter, the region experiences frequent passage of extratropical cyclones originating from the North Atlantic and Mediterranean storm tracks, bringing precipitation and colder temperatures. In contrast, summer conditions are dominated by the expansion of the subtropical height, leading to stable atmospheric conditions and minimal rainfall (ECMWF, 2010).

Synoptic-scale circulation in the Eastern Mediterranean plays a crucial role in shaping weather patterns and atmospheric dynamics. The atmospheric circulation over the eastern Mediterranean is dominated by persistent northerly and westerly winds, favoring the advection of volcanic products from Etna to Greece (Kampouri et al., 2020; Scollo et al., 2013). Research has identified several dominant synoptic types that influence the region, including cyclonic systems, anticyclonic patterns, and blocking heights (Rousi et al., 2014). These circulation patterns significantly impact the transport of aerosols, moisture, and pollutants, affecting regional air quality and climate variability. Furthermore, the region's proximity to large-scale circulation features such as the subtropical jet stream and the African monsoon system contributes to complex seasonal interactions (Lensky et al., 2018)."

**5. Page 6 Table 1 (Ash mass calculation using remote sensing data): I recommend that you add a sentence or two to justify the values selected in the table for lidar ratio and "volume to extinction conversion factor". Also, you must include some key references for that.**

**AC.** We thank the reviewer for this suggestion. We revised the text accordingly and added the following paragraph with the respective references both in the text but also included here for convenience:

The lidar ratio of coarse mode volcanic ash at 532 nm is reported to range between 40 and 60 sr in the literature (see for example Groß et al., 2012; table 3 for particle extinction and backscatter values in Floutsi et al., 2023 and Gasteiger et al., 2011). For the fine mode aerosols, we use a mean value of 60 sr following the values reported in the literature for particles of sulphuring nature (see for example Floutsi et al., 2023; Müller et al., 2007). We also account for a lidar ratio retrieval uncertainty of ~30% to capture the measurement range (Ansmann et al., 2012; Giannakaki et al., 2015; Groß et al., 2013). The particle density values  $\rho$  follow from OPAC model for coarse mode mineral component and water soluble component for ash and sulfate particles respectively (Hess et al., 1998; Koepke et al., 2015). For the water soluble component, we assume values at relative humidity of 0% which is considered representative for the altitudes of the volcanic layers. The coarse mode component is not considered as hudrophylic. Finally, the extinction to mass conversion factors cv are taken from Ansmann et al., (2011a) for ash and fine mode particles respectively.

**6. Page 12, Figure 2b (Results): It is unclear to me what meas the blue vertical lines. Profiles of NaN (no valid volume linear depolarization ratio)? Missing data in the time periods analyzed? Common time periods for lidar - photometer? Comment on this aspect.**

**AC.** We thank the reviewer for the comment. The blue vertical lines indicate negative values, which arise due to a low signal-to-noise ratio (SNR). They do not represent meaningful atmospheric features and they are masked before data averaging over the entire time interval, hence they do not influence the analysis. We added the following sentence to Figure 2 caption to make this clearer:

"The blue vertical lines on (b) indicate negative values, which arise due to a low signal-to-noise ratio (SNR) of the measurements, and they are masked before data averaging and final retrievals."

**7. Page 13, Lines 346-348: It is unclear to me why a-posterior MER has this behavior. Please give a short explanation for clarification.**

**AC.** We would like to thank the reviewer for this question. The inversion process incorporates ground-based lidar observations from the PollyXT system, which provides direct measurements of the volcanic ash plumes vertical structure. These lidar observations, being highly sensitive to the vertical distribution of ash, help refine the emission estimates, particularly within the altitude range where the plume was observed (8–12 km), ensuring that the final estimates better reflect the actual emission patterns.

Additionally, the a-priori MER estimates were initially derived from empirical relationships based on VONA-reported plume heights (see Appendix Table A1, added for further clarification). These empirical methods often overestimate emissions, particularly in the initial stages of the eruption when plume dynamics are highly variable. The inversion process corrects for this by adjusting emissions to match the lidar observations, leading to lower and more consistent a-posteriori MER values. As a result, the a-posteriori MER shows a more physically realistic and concentrated vertical distribution, predominantly between 8 and 12 km, in contrast to the a-priori estimates, which exhibit a wider spread, including lower altitudes where the plume was not actually observed. This suggests that the inversion effectively filters out unrealistic emissions, yielding a more refined and accurate vertical profile.

Moreover, it is important to highlight that during the eruption, the plume undergoes a dynamic evolution, rising and collapsing at different stages. This complex evolution is not fully captured by the model, which does not explicitly simulate the transient rise and fall of the plume. Instead, the model adjusts the a-posteriori MER at each height over time, increasing or decreasing emission rates to best match the available observations. While this approach provides a more constrained and consistent representation of emissions, some rapid fluctuations in plume height and intensity may not be fully resolved within the model framework. Finally, we emphasize that the most significant refinement in the inversion algorithm occurs between 08:00–08:45 UTC, at the peak of the eruption.

In the revised manuscript we added a paragraph to clarify better the MER behavior. The text now reads "The eruption dynamics involve a complex evolution of the volcanic plume, with phases of rising and collapsing. However, this dynamic behavior is not explicitly resolved in the a-posteriori simulations, which do not capture rapid fluctuations in plume height and intensity. Instead, the inversion algorithm adjusts the a-posteriori MER at each altitude over time, dynamically increasing or decreasing emission rates to achieve the best agreement with available observations. The most significant refinement occurs between 08:15–08:45 UTC, within the 8–12 km altitude range, where lidar observations provide direct constraints on the plume's vertical structure. As a result, the inversion optimizes the emission estimates primarily within this altitude range, ensuring the highest degree of agreement between observed and a-posteriori emissions."

8. Page 21, Lines 513-515 (Conclusions and discussion) Discuss how the method could be adjusted for other EARLINET/ACTRIS stations with similar lidar configurations and/or for the Earth Observation missions (current and future), detailing the data requirements and necessary adjustments.

**AC.** We thank the reviewer for the insightful comment. In the revised manuscript we added the following paragraph. "However, their applicability to the proposed methodology depends on the operation of a backscatter-depolarization lidar, which constitutes the primary requirement. In cases where direct measurements of essential parameters, such as lidar ratios, are unavailable, values from the scientific literature can be used. A more advanced configuration, incorporating Raman lidar capabilities, would enhance the accuracy of retrieved backscatter and lidar ratio coefficients. Additionally, for daytime measurements, a co-located sun photometer would facilitate direct estimation of the conversion factors required in the inversion process. Beyond ground-based applications, the methodology is also applicable to spaceborne aerosol lidars, which provide vertical profiles of the backscatter coefficient and particle linear depolarization ratio, both fundamental parameters for the inversion process. Furthermore, the methodology presented herein can be applied to current or future satellite missions that employ lidar measurements (e.g. the EarthCARE mission)."

**9. What were the main technical challenges in modeling long-range transported of volcanic ash and sulfate particles?**

**AC.** We would like to thank the reviewer for this question. Modeling the long-range transport of volcanic particles, especially in real time and during the initial phase of an eruption when little observational data is available (Eckhardt et al., 2008), presents several technical challenges due to their distinct physical and chemical properties of ash and sulfate particles.

Simulations of volcanic eruptions are strongly dependent on the accuracy of the eruption source term, such as 1) injection plume height, 2) the mass eruption rate (MER), 3) the vertical distribution of the ash and SO2 in the eruption column and 4) the duration of the eruption, all of which significantly influence particle dispersion (Beckett et al., 2022; Mastin et al., 2009).

Various methods exist for estimating the source term of a volcanic eruption. The most common approach is based on plume height observations, e.g., from weather radar, or ECV calibrated cameras which are then applied to empirical relationships linking the total MER to the eruption plume height (Mastin et al., 2009). However, these empirical relationships have large uncertainties, as a wide range of total mass emissions can correspond to the same plume height.

Beyond source term assumptions, there are several other factors contribute to uncertainties in volcanic particles transport modelling. These include uncertainties in the dispersion model itself, and the meteorological input data that drive the simulations. Volcanic ash and SO2 transport are highly dependent on wind fields, and errors in numerical weather prediction models, in upper-tropospheric wind shear can lead to significant uncertainties in forecasted trajectories (Harvey et al., 2020). Additionally, processes such as particle aggregation, gravitational settling, and wet/dry deposition affect ash lifetime and spatial distribution, yet they remain challenging to accurately parameterize in dispersion models (Dacre et al., 2011; Durant et al., 2010).

For volcanic ash, another important source of uncertainty is the assumptions made for particle size distribution in model simulations. Fine ash particles (diameter  $< 63 \,\mu\text{m}$ ) can remain in the atmosphere for many hours or days and can be transported far from the source, but their size distribution varies significantly between different volcanoes and eruptions (Kristiansen et al., 2012; Moxnes et al., 2014).

In the case of  $SO_2$  emissions, interactions with hydroxyl radicals (OH) in the atmosphere result in the formation of sulfate aerosols, introducing additional challenges related to chemical transformation and microphysical processes (Schmidt et al., 2012; Textor et al., 2004). Sulfate particles, due to their smaller size and longer atmospheric residence time, are prone to long-range

transport, but their interactions with clouds and solar radiation by scattering introduce further uncertainties in atmospheric modeling forecasts (Zhu et al., 2020). Addressing these challenges requires improved observational constraints, such as satellite and lidar data assimilation, to refine both ash and sulfate dispersion models (Amiridis et al., 2023; Kristiansen et al., 2010).

**10. How fast is the inversion scheme presented in this study and what is the confidence level of the data provided?**

**AC.** We would like to thank the reviewer for this question. The inversion scheme presented in this study is computationally efficient, with each simulation being processed within a few minutes. Once lidar data become available, the algorithm rapidly optimizes the volcanic ash emission profile by integrating observational constraints with dispersion modeling. This efficiency makes the method highly suitable for near-real-time applications, which are critical for aviation safety and early warning systems.

The confidence level of the provided data depends on several factors, including the accuracy of the eruption source terms (e.g., injection plume height, eruption duration), the a priori estimation of ash mass emission rates based on empirical equations, the quality of meteorological input data driving the simulations, and the constraints applied in the optimization process. Validation against ground-based lidar observations demonstrates a minimal difference of approximately 2% between the a posteriori simulated and observed ash mass concentrations. Furthermore, the assimilation of Aeolus wind data significantly enhances the accuracy of the transport model by improving wind field representation, particularly at higher altitudes. While the inversion-derived emission estimates provide a high-confidence dataset for dispersion forecasting, uncertainties persist due to potential biases in observational data and assumptions in source term estimation. To further enhance validation, additional independent datasets, such as satellite observations, ground-based remote sensing, or in situ measurements along the plume's trajectory, should be incorporated in future studies.

**Technical corrections**

**1. Update the figures 2-4 with larger fonts to make them more visible.**

AC. Done. In the revised manuscript we replace Figures 2-4.

**2. Figures 4-5: Change the "µgr" in the figure captions to "µg"**

**AC.** Done. We have changed ' $\mu$ gr' to ' $\mu$ g' in the caption of Figure 4 and updated Figure 5 accordingly. The revised caption for Figure 5 now reads: "Figure 5: FLEXPART time-height cross-sections on 12 March 2021, 18:30 – 21:30 UTC, over the PANGEA observatory in Antikythera, Greece. (a) time-height plot of a-priori FLEXPART volcanic ash concentrations ( $\mu$ g/m3), over Antikythera "w/o" Aeolus wind assimilation over Antikythera, Greece (zero values); (b) time-height plot of a-priori FLEXPART volcanic ash concentrations ( $\mu$ g/m3) over Antikythera "w" Aeolus wind assimilation; (c) time-height plot of a-posteriori FLEXPART volcanic ash concentrations ( $\mu$ g/m3) over Antikythera "w" Aeolus wind assimilation; we have updated ' $\mu$ gr' to ' $\mu$ g' in the Figure 5 plot as well.

**3. Figures 6-7: Change the "ug" in the figure captions to "µg"**

**AC.** Done. In the revised manuscript, we have changed "ug" to " $\mu$ g" in Figures 6 and 7. Additionally, as requested by **Referee 1**, we have combined Figures 6 and 7. In the revised text, Figure 7 is now presented as Figure 6 d.

**4. Please specify in the text the reference point (a.s.l or a.g.l) for the altitude values in figures 3, 6, 7 and 8.**

**AC.** Done. In the revised manuscript, we have specified the reference point (a.s.l. or a.g.l.) for the altitude values in Figures 3, 6, and 7. However, there is no Figure 8 in our manuscript.

[revised manuscript text omitted]

WMO: State of the Global Climate 2023, United Nations, Erscheinungsort nicht ermittelbar, 2024.

Zhu, Y., Toon, O. B., Jensen, E. J., Bardeen, C. G., Mills, M. J., Tolbert, M. A., Yu, P., and Woods, S.: Persisting volcanic ash particles impact stratospheric SO2 lifetime and aerosol optical properties, Nat. Commun., 11, 4526, https://doi.org/10.1038/s41467-020-18352-5, 2020.

---

## Author Comment (AC3)

**AC answer to CC1**

We would like to thank Mr. S. Singh for his comments. Our replies (regular font) for each comment (bold font) are provided below.

**- The use of the inversion algorithm in this manner appears highly problematic and ad hoc. For instance, if equation (5) holds, then the shapes in equation (7) do not match.**

**AC.** We thank Mr. S. Singh for his valuable comments. The inversion methodology presented in this study is based on cost function minimization, following a Bayesian approach, as established in previous studies (Eckhardt et al., 2008; Stohl, 2011; Kristiansen et al., 2014). Additionally, the inversion scheme developed in this study combines a-priori source information with the output of the FLEXPART dispersion model and PollyXT lidar retrievals to estimate volcanic ash emissions more accurately, ensuring that the approach is well-structured and methodologically robust. A key innovation of this study is the integration of ground-based data into the inversion scheme.

Upon carefully reviewing our calculations, we confirm that the matrix M (SRM) has correct shape of (nxm) and not (mxn), as previously stated in the text. We acknowledge this typographical mistake and have corrected it in the revised manuscript. With this correction, the shapes in Equation (5) and Equation (7) are consistent, ensuring the mathematical coherence of the formulation.

**Furthermore, it is implicitly assumed in (7)-(9) that uncertainties associated with all observations are equal to 1. I do not believe this assumption to be valid.**

**AC.** Thank you for your valuable comment. The assumption that all uncertainties associated with the observations are equal to 1 is made because uncertainty is not explicitly addressed in this work. However, we acknowledge that the uncertainties should be included in the inversion calculations and plan to incorporate them in our future work. We will make this clearer in the revised manuscript.

**The absence of a regularization parameter in term (8) suggests that the regularization is implicitly taken as 1, but I find it difficult to believe that this is optimal. The authors do not provide any optimization scheme for the regularization parameter in equation (8) or any explanation of how epsilon has been selected.**

**AC.** In this study the regularization parameter was found using a grid search. However, the regularization parameter $\epsilon$ is used to enforce smoothness in the vertical profile of emissions and is derived from a discrete second-order difference operator. While a formal optimization of $\epsilon$ is not performed, sensitivity experiments have been conducted to assess its influence on the solution, ensuring that the chosen value provides a stable and reliable inversion outcome.

**-Additionally, no sensitivity study to a-priori emissions has been presented.**

**AC.** Thank you for your comment. At this stage of the study, we did not perform a sensitivity study on a-priori emissions. However, we acknowledge its importance and plan to incorporate such an analysis in future work.

**- I strongly recommend that the inversion algorithm be made available in a public repository for scrutiny.**

**AC.** While the inversion methodology is detailed in the paper, the specific algorithm implementation is not available online. However, the code is available by the authors upon request.

**- Since the authors have SEVIRI data, they should also perform inversion for SEVIRI data and compare the results derived from lidar and SEVIRI (In fact the same team performed the same study already and presented the work at EGU: https://meetingorganizer.copernicus.org/EGU23/EGU23-13755.html). This would serve as a basic form of validation, as has also been pointed out by respected Reviewer 2. At present, there is no validation of the results. However, considering the issues raised above, the publication of this study must be considered with utmost caution.**

**AC.** We thank him for this suggestion. We acknowledge the importance of performing an inversion using SEVIRI data and comparing the results with those derived from lidar observations, as this would indeed serve as a valuable validation step. This aspect is already in our plans for future work, and we are currently preparing a follow-up study that will focus on this analysis in more detail.

In the present version of our work, we are primarily focused on demonstrating an inversion scheme using ground-based observations. Our goal is to establish the methodology and assess its effectiveness before extending the analysis to satellite-based retrievals. We appreciate the reference to our previous work presented at EGU and fully agree that integrating SEVIRI data along with observations from other satellite missions such as EarthCARE, would further enhance the robustness of our findings.

---

## Author Response (AR2)

**AC to Editor comment,**

We would like to thank the Editor's and Dr. Singh's comments regarding the uncertainty about the estimated emissions. AC reply (regular font) for the comment (bold font) is provided below.

**I think one point that have to be addressed a bit more is based on the comments of Dr. Singh on the uncertainty related with the output (measurements/model). I think that in order to demonstrate the method usability for different applications some brief discussion on the at least the range of related uncertainties could be added in the conclusion section. I think this will enhance the paper value.**

To quantify the uncertainty in our emission estimates, we applied a Monte Carlo error propagation approach. At each height level, we perturbed the lidar profile $y_o$ by adding normally distributed noise, drawn from $N(0, \sigma^2)$, where $\sigma$ is the standard error associated with each measurement. By repeating this process over multiple iterations, we obtained a set of emission profiles, allowing us to compute the mean and standard deviation at each height level. This method provides an estimate of the uncertainty in the retrieved emissions.

However, the results indicate that the inversion output remains highly stable, with minimal variation across Monte Carlo realizations. This suggests that, in the current setup, the available observational constraints based on a single lidar station do not introduce significant uncertainty into the retrieved emissions. This stability highlights the need for additional observational constraints to better validate the results.

Having multiple lidar stations would significantly improve the reliability of the retrieved emissions by offering a more comprehensive observational dataset. However, for this specific case study, additional ground-based observations were not available. For future research, we recommend incorporating multiple observation sites or complementary remote sensing techniques to enhance the sensitivity of the inversion framework and achieve a more thorough uncertainty assessment.

In the revised text of the Conclusions and Discussions section, we have added the following paragraph. The text now reads:

"To further assess the reliability of the retrieved emissions, a Monte Carlo error propagation analysis was conducted, introducing normally distributed perturbations to the lidar measurements. With this method the standard deviation of the retrieved emissions at each height level was estimated. The results indicate that the inversion output remained highly stable, with minimal variation across Monte Carlo realizations, suggesting that the single-station observational setup does not introduce significant uncertainty. To enhance the sensitivity of the inversion framework and provide a more comprehensive uncertainty assessment, multiple lidar stations or complementary remote sensing techniques are essential."